# DISCRETE DISTRIBUTION NETWORKS

**Lei Yang**
StepFun
Megvii Technology
yanglei@stepfun.com

## ABSTRACT

We introduce a novel generative model, the Discrete Distribution Networks (DDN), that approximates data distribution using hierarchical discrete distributions. We posit that since the features within a network inherently capture distributional information, enabling the network to generate multiple samples simultaneously, rather than a single output, may offer an effective way to represent distributions. Therefore, DDN fits the target distribution, including continuous ones, by generating multiple discrete sample points. To capture finer details of the target data, DDN selects the output that is closest to the Ground Truth (GT) from the coarse results generated in the first layer. This selected output is then fed back into the network as a condition for the second layer, thereby generating new outputs more similar to the GT. As the number of DDN layers increases, the representational space of the outputs expands exponentially, and the generated samples become increasingly similar to the GT. This hierarchical output pattern of discrete distributions endows DDN with unique properties: more general zero-shot conditional generation and 1D latent representation. We demonstrate the efficacy of DDN and its intriguing properties through experiments on CIFAR-10 and FFHQ. The code is available at `https://discrete-distribution-networks.github.io/`

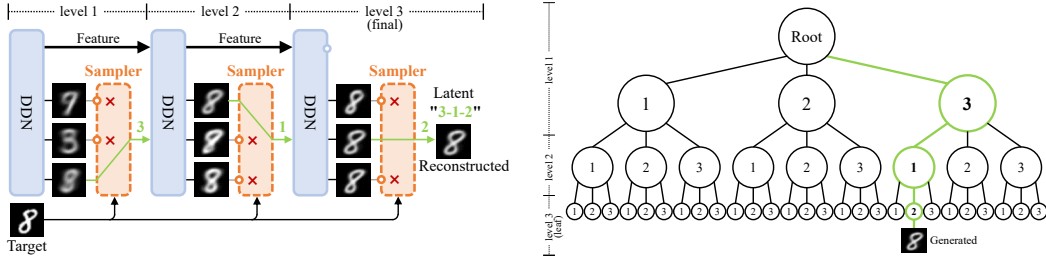

(a) Image Reconstruction through DDN      (b) Tree Structure of DDN's Latent

Figure 1: (a) Illustrates the process of image reconstruction and latent acquisition in DDN. Each layer of DDN outputs $K$ distinct images to approximate the distribution $P(X)$. The sampler then selects the image most similar to the target from these and feeds it into the next DDN layer. As the number of layers increases, the generated images become increasingly similar to the target. For generation tasks, the sampler is simply replaced with a random choice operation. (b) Depicts the tree-structured representation space of DDN's latent variables. Each sample can be mapped to a leaf node on this tree.

## 1 INTRODUCTION

With the advent of ChatGPT Brown et al. (2020) and DDPM Ho et al. (2020), deep generative models have become increasingly popular and significant in everyday life. However, modeling the complex and diverse high-dimensional data distributions is challenging. Previous methods Kingma & Welling (2014); Radford et al. (2016); Kingma & Dhariwal (2018); Goyal et al. (2021); Song et al. (2021); Shocher et al. (2024); Graves et al. (2023) have each demonstrated their unique strengths and characteristics in modeling these distributions. In this work, we propose a novel approach to model

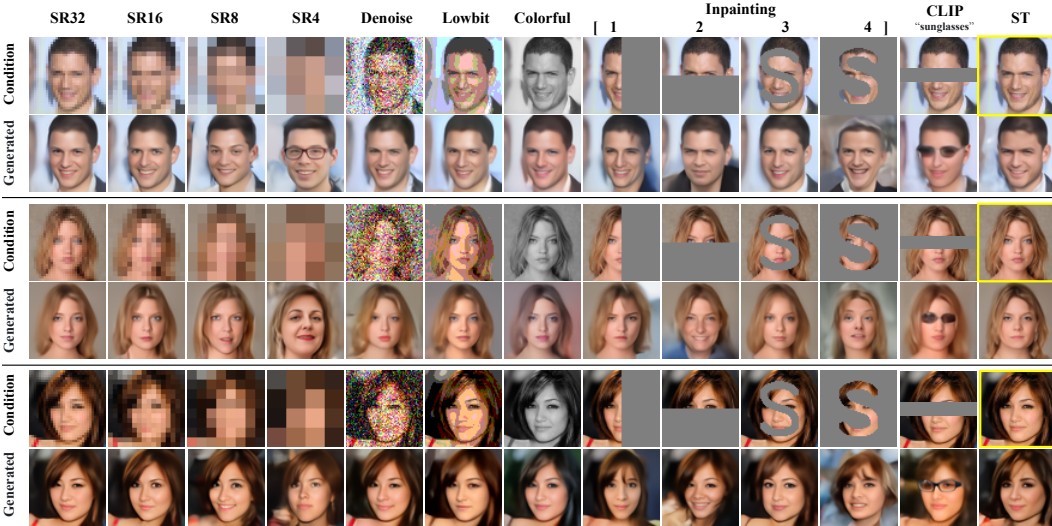

Figure 2: **DDN enables more general zero-shot conditional generation.** DDN supports zero-shot conditional generation across non-pixel domains, and notably, without relying on gradient, such as text-to-image generation using a black-box CLIP model Radford et al. (2021). Images enclosed in yellow borders serve as the ground truth. The abbreviations in the table header correspond to their respective tasks as follows: 'SR' stands for Super-Resolution, with the following digit indicating the resolution of the condition. 'ST' denotes Style Transfer, which computes Perceptual Losses with the condition according to Johnson et al. (2016).

the target distribution, where the core idea is to generate multiple samples simultaneously, allowing the network to directly output an approximate discrete distribution. Hence, we name our method Discrete Distribution Networks (DDN). DDN embraces a core concept as simple as autoregressive models, offering another straightforward and effective form for generative models.

Most generative models applied in real-world scenarios are conditional generative models. Taking image generation as an example, these models generate corresponding images based on content provided by users, such as images to be edited, reference images Saharia et al. (2021), text descriptions, hand-drawn editing strokes Voynov et al. (2022), sketches, and so on. Current mainstream generative models Rombach et al. (2021); Ramesh et al. (2022); Zhang et al. (2023) typically require training separate models and parameters for each condition. These models are restricted to fixed condition formats and lack the flexibility to adjust the influence of each condition dynamically, thereby limiting users' creative freedom.

Recent works have attempted to address this issue through zero-shot conditional generation (ZSCG). However, these methods either only support conditions in the same pixel domain as the training data Wang et al. (2022); Lugmayr et al. (2022); Meng et al. (2021); Nair et al. (2023) or depend on discriminative models to supply gradients during generation Yu et al. (2023). In contrast, DDN supports a wide range of ZSCG tasks, encompassing both pixel-domain and non-pixel-domain conditions, as shown in fig. 2. To the best of our knowledge, DDN is the first generative model capable of performing zero-shot conditional generation in non-pixel domains without relying on gradient information. This implies that DDN can achieve ZSCG solely based on black-box discriminative models.

The core concept of Discrete Distribution Networks (DDN) is to approximate the distribution of training data using a multitude of discrete sample points. The secret to generating diverse samples lies in the network's ability to concurrently generate multiple samples ($K$). This is perceived as the network outputting a discrete distribution. All generated samples serve as the sample space for this discrete distribution. Typically, each sample in this discrete distribution has an equal probability mass of $1/K$. Our goal is to make this discrete distribution as close as possible to the target dataset.

To accurately fit the target distribution of large datasets, a substantial representational space is required. In the most extreme scenario, this space must be larger than the number of training data samples. However, current neural networks lack the feasibility to generate such a vast number of

samples simultaneously. Therefore, we adopt a strategy from autoregressive models Van Den Oord et al. (2016) and partition this large space into a hierarchical conditional probability model. Each layer of this model needs only a small number of outputs. We then select one of these outputs as the output for that layer and use it as conditional input to the next layer. As a result, the output of the next layer will be more closely related to the selected conditional sample. If the number of layers is $L$ and the number of outputs per layer is $K$, then the output space of the network is $K^L$. Due to its exponential nature, this output space will be much larger than the number of samples in the dataset. Figure 1 shows how DDN generates images.

We posit that the contributions of this paper are as follows:

- We introduce a novel generative model, termed Discrete Distribution Networks (DDN), which exhibit a more straightforward and streamlined principle and form.

- For training the DDN, we propose the "Split-and-Prune" optimization algorithm, and a range of practical techniques.

- We conduct preliminary experiments and analysis on the DDN, showcasing its intriguing properties and capabilities, such as zero-shot conditional generation and 1D latent representations.

## 2 RELATED WORK

**Deep Generative Model.** Generative Adversarial Networks (GANs) Radford et al. (2016); Brock et al. (2019) and Variational Autoencoders (VAEs) Kingma & Welling (2014); van den Oord et al. (2018) are two early successful generative models. GANs reduce the divergence between the generated sample distribution and the target distribution through a game between the generator and discriminator. However, regular GANs cannot map samples back to the latent space, thus they cannot reconstruct samples. VAEs encode data into a simple distribution's latent space through an Encoder, and the Decoder is trained to reconstruct the original data from this simple distribution's latent space. Autoregressive models van den Oord et al. (2016) , with their simple principles and methods, model the target distribution by decomposing the target data into conditional probability distributions of each component. They can also compute the exact likelihood of target samples. However, the efficiency of these models is reduced when dealing with image data, which is not suitable to be decomposed into a sequence of components. Normalizing flow Kingma & Dhariwal (2018) is another class of generative models that can compute the likelihood. They use invertible networks to construct a mapping from samples to a noise space, and during the generation stage, they map back from noise to samples. Energy Based Models (EBM) Goyal et al. (2021); Song et al. (2021) with their high-quality and rich generative results, have led to the rise of diffusion models. However, their multi-step iterative generation process requires substantial computational resources. The recent Idempotent Generative Network Shocher et al. (2024) introduces a novel approach by training a neural network to be idempotent, mapping any input to the target distribution effectively.

**Connections to VQ-VAE**. While both VQ-VAE van den Oord et al. (2018) and DDN involve discrete representations, they differ significantly in their approach and capabilities. VQ-VAE enhances the traditional VAE by replacing the continuous latent space with discrete codebooks, thus achieving a more compact representation. VQ-VAE-2 Razavi et al. (2019) further improves this by employing a multi-scale hierarchical structure, thereby enhancing its representational power. However, the discrete representation in VQ-VAE remains two-dimensional, potentially leading to redundant information. Furthermore, VQ-VAE and its successors still rely on an additional prior network for generative modeling in the latent space. Notably, DDN can also serve as this prior model to effectively model the latent space of VQ-VAE. Other distinctions between DDN and VQ-VAE include the absence of an encoder and codebook in DDN, as well as its capacity for Zero-Shot Conditional Generation. VQ-VAEs are known to encounter codebook collapse, a problem that some researchers have addressed by reinitializing unused codes near frequently used ones Williams et al. (2020); Dhariwal et al. (2020). Our Split-and-Prune algorithm shares a similar core idea, albeit with some differences. While the reinitialization method aims to balance code usage to mitigate codebook collapse, our goal is to align the discrete distribution output by our network as closely as possible to the target distribution.

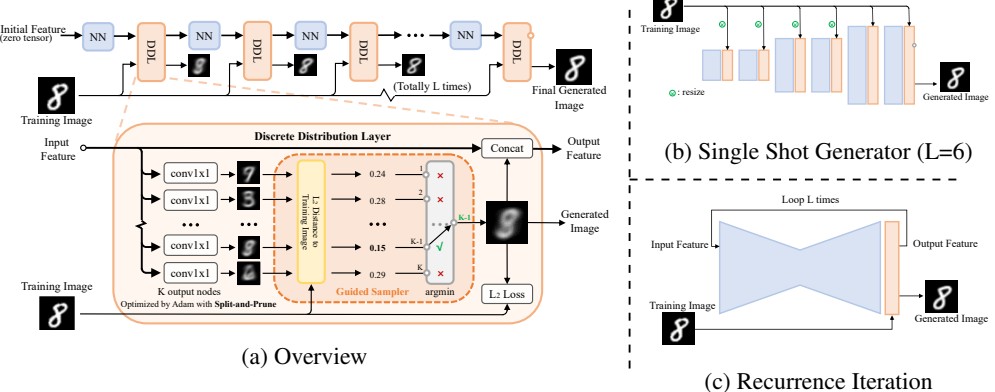

(a) Overview

(b) Single Shot Generator (L=6)

(c) Recurrence Iteration

Figure 3: **Schematic of Discrete Distribution Networks (DDN).** (a) The data flow during the training phase of DDN is shown at the top. As the network depth increases, the generated images become increasingly similar to the training images. Within each Discrete Distribution Layer (DDL), $K$ samples are generated, and the one closest to the training sample is selected as the generated image for loss computation. These $K$ output nodes are optimized using Adam with the Split-and-Prune method. The right two figures shown the two model paradigms supported by DDN. (b) Single Shot Generator Paradigm: Each neural network layer and DDL has independent weights. (c) Recurrence Iteration Paradigm: All neural network layers and DDLs share weights. For inference, replacing the Guided Sampler in the DDL with a random choice enables the generation of new images.

## 3  DISCRETE DISTRIBUTION NETWORKS

**Network architecture.** Figure 3a illustrates the overall structure of the DDN, comprised of Neural Network Blocks and Discrete Distribution Layers (DDL). Each DDL contains $K$ output nodes, each of which is a set of 1x1 convolutions responsible for transforming the feature into the corresponding output image. The parameters of these 1x1 convolutions are optimized by Adam with Split-and-Prune. The $K$ images generated by the $K$ output nodes are inputted into the Guided Sampler. The Guided Sampler selects the image with the smallest L2 distance to the training image, which serves as the output of the current layer and is used to calculate the L2 loss with the training image. Simultaneously, the selected image is concatenated back into the feature, acting as the condition for the next block. The index (depicted in green as "**K-1**" in fig. 3a) of the selected image represents the latent value of the training sample at this layer. Through the guidance of the Guided Sampler layer by layer, the image generated by the network progressively becomes more similar to the training sample until the final layer produces an approximation of the training sample.

For computational efficiency, we adopted a decoder structure similar to the generator in GANs for coarse-to-fine image generation, as shown in fig. 3b. We refer to this as the Single Shot Generator which is our default choice. As each layer of DDN can naturally input and output RGB domain data, DDN seamlessly support the Recurrence Iteration Paradigm fig. 3c.

**Objective function.** The DDN model consists of $L$ layers of Discrete Distribution Layers (DDL). For a given layer $l$, denoted as $f_l$, the input is the selected sample from the previous layer, $\mathbf{x}_{l-1}^*$. The layer generates $K$ new samples, $f_l(\mathbf{x}_{l-1}^*)$, from which we select the sample $\mathbf{x}_l^*$ that is closest to the current training sample $\mathbf{x}$, along with its corresponding index $k_l^*$. The loss $J_l$ for this layer is then computed only on the selected sample $\mathbf{x}_l^*$.

$$k_l^* = \operatorname*{argmin}_{k \in \{1,\dots,K\}} \left\| f_l(\mathbf{x}_{l-1}^*)[k] - \mathbf{x} \right\|^2 \tag{1}$$

$$\mathbf{x}_l^* = f_l(\mathbf{x}_{l-1}^*)[k_l^*] \quad ; \quad J_l = \left\| \mathbf{x}_l^* - \mathbf{x} \right\|^2 \tag{2}$$

Here, $\mathbf{x}_0^* = \mathbf{0}$ represents the initial input to the first layer. For simplicity, we omit the details of input/output feature, neural network blocks and transformation operations in the equations.

By recursively unfolding the above equations, we can derive the latent variable $\mathbf{k}_{1:L}^*$ and the global objective function $J$.

$$\mathbf{k}_{1:L}^* = [k_1^*, k_2^*, \ldots, k_L^*] = \left[ \underset{k \in \{1,\ldots,K\}}{\arg\min} \left\| \mathcal{F}([\mathbf{k}_{1:l-1}^*, k]) - \mathbf{x} \right\|^2 \right]_{l=1}^{L} \tag{3}$$

$$J = \frac{1}{L} \sum_{l=1}^{L} \left\| \mathcal{F}(\mathbf{k}_{1:l}^*) - \mathbf{x} \right\|^2 \tag{4}$$

Here, $\mathcal{F}$ represents the composite function formed from $f_l$, defined as: $\mathcal{F}(\mathbf{k}_{1:l}) = f_l(f_{l-1}(\ldots f_1(\mathbf{x}_0)[k_1]\ldots)[k_{l-1}])[k_l]$. Finally, we average the L2 loss across all layers to obtain the final loss for the entire network.

**How to generation.** When the network performs the generation task, replacing the Guided Sampler with a random choice enables image generation. Given the exponential representational space of $K^L$ sample points and the limited number of samples in the training set, the probability of sampling an image with the same latent space as those in the training set is also exponentially low. For image reconstruction tasks, the process is almost identical to the training process, only substituting the training image with the target image to be reconstructed and omitting the L2 Loss part from the training process. The Final Generated Image in fig. 3a represents the final reconstruction result. The indices of the selected samples along the way form the target image's latent $\mathbf{k}_{1:L}^*$, same as eq. (3). Therefore, the latent $\mathbf{k}_{1:L}^*$ is a sequence of integers with length $L$, which we regard as the hierarchical discrete representation of the target sample. The latent space exhibits a tree structure with $L$ layers and $K$ degrees per node, where each leaf node represents a sample point, and its latent denotes the indices of all nodes along the path to this leaf node, as shown in fig. 1b. In the latent sequence, values placed earlier correspond to higher-level nodes in the tree, controlling the low-frequency information of the output sample, while later values tend to affect high-frequency information.

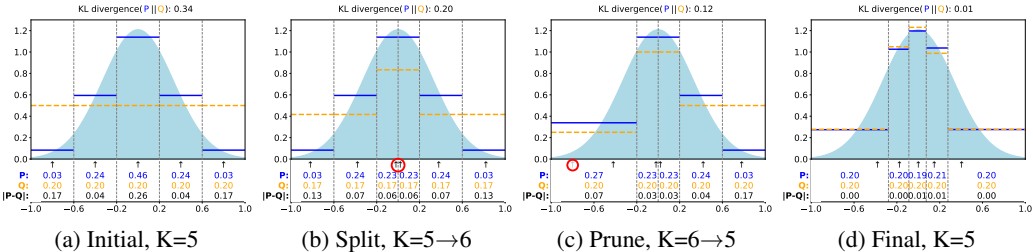

(a) Initial, K=5  (b) Split, K=5→6  (c) Prune, K=6→5  (d) Final, K=5

Figure 4: **Illustration of the principle behind the Split-and-Prune operation.** For example in (a), the light blue bell-shaped curve represents a one-dimensional target distribution. The 5 "↑" under the x-axis are the initial values from a uniform distribution of 5 output nodes, which divide the entire space into 5 parts using midpoints between adjacent nodes as boundaries (i.e., vertical gray dashed lines). Each part corresponds to the range represented by this output node on the continuous space $x$. Below each node are three values: $P$ stands for the relative frequency of the ground truth falling within this node's range during training; $Q$ refers to the probability mass of this sample (node) in the discrete distribution output by the model during the generation phase, which is generally equal for each sample, i.e., $1/K$. The bottom-most value denotes the difference between $P$ and $Q$. Colorful horizontal line segments represent the average probability density of $P$, $Q$ within corresponding intervals. In (b), the Split operation selects the node with the highest $P$ (circled in red). In (c), the Prune operation selects the node with the smallest $P$ (circled in red). In (d), through the combined effects of loss and Split-and-Prune operations, the distribution of output nodes moves towards final optimization. From the observed results, the KL divergence ($KL(P||Q)$) consistently decreases as the operation progresses, and the yellow line increasingly approximates the light blue target distribution.

## 3.1 OPTIMIZATION WITH SPLIT-AND-PRUNE

We have observed two primary issues resulting from each layer calculating loss only with the closest output samples to the ground truth (GT). Firstly, a problem similar to the "dead codebooks" in VQ-VAE, wherein output nodes that are not selected for a long time receive no gradient updates. During generation, these "dead nodes" are selected with equal probability, leading to poor output. The second issue is the probability density shift. For instance, in a one-dimensional asymmetric bimodal distribution, the target distribution is a mixture of two thin and tall Gaussian distribution functions, with one larger and one smaller peak. The means of these two peaks are -1 and 1, respectively. Therefore, half of the output samples with initial values less than 0 will be matched with samples from the larger peak and optimized towards the larger Gaussian distribution. Meanwhile, the other half of the output samples with values greater than 0 will be optimized towards the smaller Gaussian. A problem arises as the large and small peaks carry different probability masses but occupy equal portions of the output samples, resulting in the same sampling probability during generation.

Inspired by the theory of evolution and genetic algorithms Katoch et al. (2021), we propose the Split-and-Prune algorithm to address the above issues, as outlined in algorithm 1. The Split operation targets nodes frequently matched by training samples, while the Prune operation addresses the issue of "dead nodes", similar to those in Williams et al. (2020); Dhariwal et al. (2020). These nodes are akin to species in evolution, subject to diversification and extinction. During training, we track the frequency with which each node is matched by training samples. For nodes with excessive frequency, we execute the Split operation, cloning the node into two new nodes, each inheriting half of the old node's frequency. Although these two new nodes have identical parameters and outputs, the next matched training sample will only be associated with one node. Therefore, the loss and gradient only affect one node's parameters. Consequently, their parameters and outputs exhibit slight differences, dividing the old node's match space into two. In subsequent training, the outputs of the two new nodes will move towards the centers of their respective spaces under the influence of the loss, diverging to produce more diverse outputs. For nodes with low matching frequency (dead nodes), we implement the Prune operation, removing them outright. Figure 4 illustrates the process of the Split-and-Prune operation and how it reduces the distance between

---

**Algorithm 1** Split-and-Prune of one layer

---

**Require:** Output nodes number $K$, model $f$, non-output parameters $\boldsymbol{\theta}$, target distribution $q(\mathbf{x})$
1: Initialize output node parameters $\boldsymbol{\psi}(k)$ for $k \in \{1, \ldots, K\}$ with random values
2: Initialize counter $\mathbf{c}(k) = 0$ for $k \in \{1, \ldots, K\}$
3: Set split/prune threshold $P_{split} \leftarrow 2/K$, $P_{prune} \leftarrow 0.5/K$
4: $n \leftarrow 0, k_{new} \leftarrow K + 1$
5: **repeat**
6:     $\mathbf{x} \sim q(\mathbf{x})$
7:     Choose $k^* = \underset{k \in \boldsymbol{\psi}}{\operatorname{argmin}} \|f(\boldsymbol{\theta}, \boldsymbol{\psi}(k)) - \mathbf{x}\|^2$
8:     Gradient descent $\nabla_{\boldsymbol{\theta},\boldsymbol{\psi}(k^*)} \|f(\boldsymbol{\theta}, \boldsymbol{\psi}(k^*)) - \mathbf{x}\|^2$
9:     $\mathbf{c}(k^*) := \mathbf{c}(k^*) + 1$
10:     $n \leftarrow n + 1$
11:     $k_{max} = \underset{k}{\operatorname{argmax}} \mathbf{c}(k)$ and $k_{min} = \underset{k}{\operatorname{argmin}} \mathbf{c}(k)$
12:     **if** $\mathbf{c}(k_{max})/n > P_{split}$ or $\mathbf{c}(k_{min})/n < P_{prune}$ **then**
        # Split:
13:         $\boldsymbol{\psi}(k_{new}) := \operatorname{clone}(\boldsymbol{\psi}(k_{max}))$
14:         $\mathbf{c}(k_{new}) := \mathbf{c}(k_{max})/2$
15:         $\mathbf{c}(k_{max}) := \mathbf{c}(k_{max})/2$
16:         $k_{new} \leftarrow k_{new} + 1$
        # Prune:
17:         $n \leftarrow n - \mathbf{c}(k_{min})$
18:         remove $\boldsymbol{\psi}(k_{min})$ and $\mathbf{c}(k_{min})$
19:     **end if**
20: **until** converged

---

the discrete distribution represented by the Output Nodes and the target distribution. The efficacy of the Split-and-Prune optimization algorithm is validated through examples of fitting 2D density maps in fig. 17.

## 3.2 APPLICATIONS

**Zero-Shot Conditional Generation (ZSCG).** The DDN sampling process occurs directly within the sample space, which inherently facilitates Zero-Shot Conditional Generation. Each layer of the DDN generates multiple target samples, with a selected sample being forwarded as a condition to the next layer. This enables the generation of new samples in the desired direction, ultimately producing a sample that meets the given condition. In fact, the reconstruction process shown in

fig. 1a is a ZSCG process guided by a target image. It is important to note that the target image is never directly input into the network. Instead, it shapes the generation outcomes by steering the sampling process.

To implement ZSCG, we replace the Guided Sampler in fig. 3a with a Conditional Guided Sampler. For instance, when generating an image of class $y_i$ guided by a classifier $g_{cls}$, we replace the "L2 Distance to Training Image" in the Guided Sampler of fig. 3a with the probability of each output image belonging to class $y_i$ according to the classifier. We then replace "argmin" with "argmax" to construct the Guided Sampler for this classifier. Similar to eq. (1), the sampling method is:

$$k_l^* = \operatorname*{argmax}_{k \in \{1,\ldots,K\}} g_{cls}(f_l(\mathbf{x}_{l-1}^*)[k])[y_i] \tag{5}$$

After performing $L$ steps guided sampling and $L \times K$ steps of classification, the ZSCG result can be obtained without any gradient.

For super-resolution and colorization tasks, we construct a transform that converts the generated images into the target domain (low-resolution or grayscale). This approach significantly reduces the impact of the missing signal from the condition on the generated images, enabling DDN to successfully perform super-resolution tasks even when the source image has a resolution as low as $4 \times 4$.

The use of "argmax" in the Guided Sampler causes each layer to select a fixed sample, resulting in the same image output under identical conditions, similar to greedy sampling in GPT. To increase diversity, we use Top-k sampling with $k = 2$ for most ZSCG tasks, which balances diversity and condition appropriateness within the large generation space $(2^L)$.

The versatility of ZSCG can be further enhanced by combining different Guided Samplers. For example, an image can guide the primary structure while text guides the attributes. The influence of each condition can be adjusted by setting their respective weights. Experiments involving the combination of different Guided Samplers are presented in appendix B.

**Efficient Data Compression Capability.** The latent of DDN is a highly compressed discrete representation, where the information content of a DDN latent is $L \times \log_2 K$ bits. Taking our default experimental values of $K$=512 and $L$=128 as an example, a sample can be compressed to 1152 bits, demonstrating the efficient lossy compression capability of DDN. We hypothesize this ability originates from two aspects: 1) the compact hierarchical discrete representation, and 2) the Split-and-Prune operation makes the probabilities of each node as equal as possible, thereby increasing the information entropy Shannon (1948) of the entire latent distribution and more effectively utilising each bit within the latent.

In our experiments, we set $K$=512 as the default, considering the balance between generation performance and training efficiency. However, from the perspective of data compression, setting $K$ to 2 and increasing $L$ provides a better balance between representation space and compression efficiency. We refer to DDN with $K = 2$ as Taiji-DDN because of its similarity to the concept of Taiji in ancient Chinese philosophy, as described in fig. 18. To our knowledge, Taiji-DDN is the first generative model capable of directly transforming data into a semantically meaningful binary string which represents a leaf node on a balanced binary tree.

## 3.3 TECHNIQUES

In this subsection, we present several techniques for training Discrete Distribution Networks.

**Chain Dropout.** In scenarios where the number of training samples is limited, each data sample undergoes multiple training iterations within the network. During these iterations, similar selections are often made by the Guided Sampler at each layer. However, the representational space of DDN far exceeds the number of training samples in the dataset. This disparity leads to a situation where the network is only trained on a very limited set of pathways, resulting in what can be perceived as overfitting on these pathways. To mitigate this, we introduce a strategy during training where each Discrete Distribution Layer substitutes the Guided Sampler with a "random choice" at a fixed probability rate. We refer to this method as "Chain Dropout".

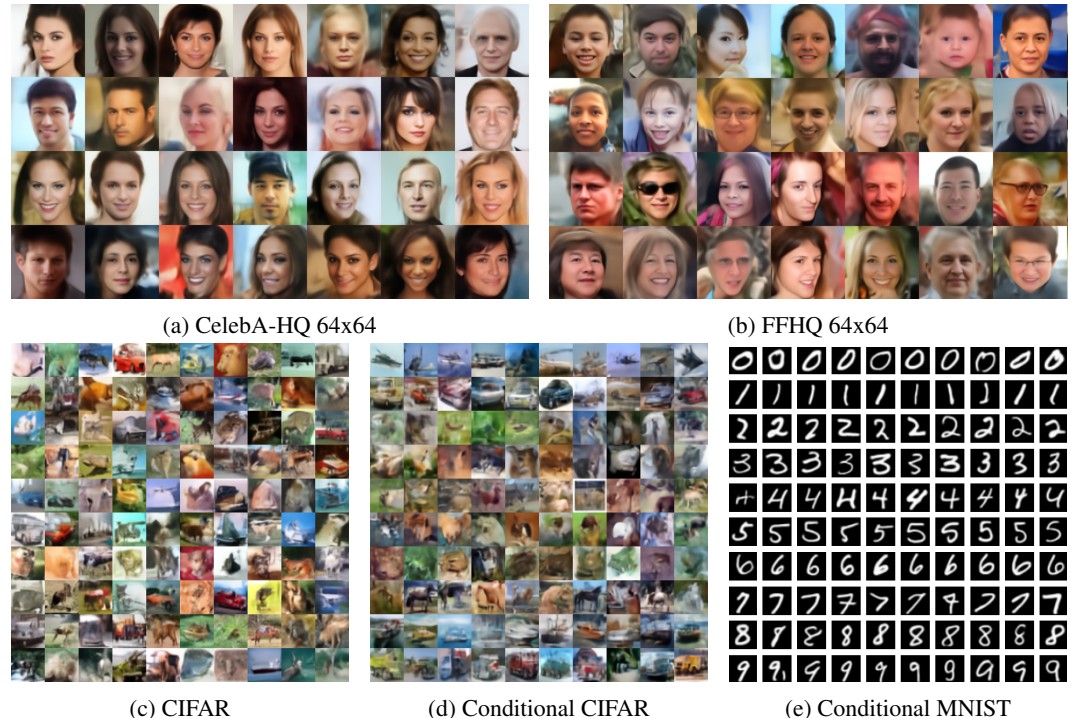

(a) CelebA-HQ 64x64            (b) FFHQ 64x64

(c) CIFAR        (d) Conditional CIFAR       (e) Conditional MNIST

Figure 5: **Random samples from DDN.** Figures (d) and (e) showcase images that are conditionally generated by conditional DDN, with each row of images representing a distinct category.

**Learning Residual.** In the context of utilizing the Single Shot Generator structure, a Discrete Distribution Layer is introduced every two convolution layers. Given such small amount of computation between adjacent layers, directly regressing the images themselves with these convolutions becomes challenging. Drawing inspiration from ResNet He et al. (2016), we propose a scheme for the network to learn the residual between the output images from the preceding layer and the ground truth. The output of the current layer is then computed as the sum of the output from the previous layer and the current layer's residual. This approach alleviates the pressure of the network to represent complex data and enhances the flexibility of the network.

**Leak Choice.** In each DDN layer, the output is conditioned on the image selected from the previous layer. This condition serves as a signal in the image domain, requiring the current layer to expend computational resources to extract features and interpret the choices made by the sampler. However, the computation between two adjacent layers in Single Shot Generator Paradigm is very small, involving only two convolutional layers. To facilitate a faster understanding of the choices made by the Sampler in the subsequent layer, we have added extra convolutional layers to each output node. The features produced by these extra convolutions also serve as conditional inputs to the next layer. But these features don't participate in the calculation of distance or loss in DDL.

## 4 EXPERIMENTS

We trained our models on a server equipped with 8 RTX2080Ti GPUs, setting the Chain Dropout probability to 0.05 by default. For the 64x64 resolution experiments, we utilized a DDN with 93M parameters, setting $K = 512$ and $L = 128$. In the CIFAR experiments, we employed a DDN with 74M parameters, setting $K = 64$ and $L = 64$. The MNIST experiments were conducted using a Recurrence Iteration Paradigm UNet model with 407K parameters, where $K = 64$ and $L = 10$. DDN is implemented on the foundation of the EDM Karras et al. (2022) codebase, with training parameters nearly identical to EDM. More extended experiments exploring the properties of DDN can be found in the appendix.

## 4.1 QUALITATIVE AND QUANTITATIVE RESULTS

**Generation Quality.** Figures 5a and 5b depict random generation results of DDN on CelebA-HQ-64x64 Karras et al. (2017) and FFHQ-64x64 Karras et al. (2019), verifying the model's effectiveness in modeling facial data. The generation quality on CelebA-HQ appears superior to that on FFHQ, which is also reflected in the lower FID score (35.4 VS 43.1). We surmise this disparity arises from CelebA-HQ's relatively cleaner backgrounds and less diverse facial data compared to FFHQ.

Figure 5c showcases the random generation results of DDN on the CIFAR dataset. We also present the FID score of DDN on unconditional CIFAR in table 1, comparing it with classical generative models. It is worth noting that modeling CIFAR remains a challenging task, especially for new and under-explored generative models like DDN. For instance, the recent work IGN Shocher et al. (2024) did not conduct experiments on CIFAR.

Table 1: **Quantitative comparison on CIFAR-10.** The data for VQ-VAE comes from Vuong et al. (2023). Data for other baselines comes from Bond-Taylor et al. (2021).

| Method | Type | FID↓ |
|--------|------|------|
| DCGAN | GAN | **37.1** |
| IGEBM | EBM | 38.0 |
| VAE | VAE | 106.7 |
| VQ-VAE | VAE | 117.4 |
| Gated PixelCNN | AR | 65.9 |
| GLOW | Flow | 46.0 |
| DDN(ours) | DDN | 52.0 |

**Conditional Training.** Training a conditional DDN is quite straightforward, it only requires the input of the condition or features of the condition into the network, and the network will automatically learn $P(X|Y)$. Figures 5d and 5e show the generation results of the class-conditional DDN on CIFAR and MNIST, respectively. Figure 8 further demonstrates the combination of conditional generation and ZSCG on image-to-image tasks.

**Zero-Shot Conditional Generation.** We trained a DDN model on the FFHQ-64x64 dataset and then evaluated its zero-shot conditional image generation capability using the CelebA-HQ-64x64 dataset, as shown in fig. 2. In appendix B, we presented the experiments of text-guided ZSCG using CLIP Radford et al. (2021). In particular, our generation process does not require gradient derivation, numerical optimization, or iterative steps. To the best of our knowledge, DDN is the first generative model to support the use of a purely discriminative model as a guide for zero-shot conditional generation.

In addition, we employed an off-the-shelf CIFAR classifier Phan (2021) to guide the generation of specific category images by a DDN model trained unconditionally on CIFAR. We want to emphasize that the classifier is an open-source, pre-trained ResNet18 model, with no additional modifications or retraining. Figure 7 displays images of various CIFAR categories generated by the model under the guidance of the classifier.

Table 2: **Ablation study on FFHQ-64x64.** We use the reconstruction Fréchet Inception Distance (rFID) to reflect the reconstructive performance of the network. All models are trained on the FFHQ-64x64 dataset. The rFID-FFHQ represents the reconstructive performance of the model on the training set, while rFID-CelebA can be seen as an indication of the model's generalization performance on the test set. "w/o" stands for "without".

**Latent analysis.** We trained a DDN on the MNIST dataset with $K = 8$ and $L = 3$ to visualize both the hierarchical generative behavior of the DDN and the distribution of samples in the entire latent representation space, as shown in fig. 6.

| Model | FID↓ | FFHQ↓ | CelebA↓ |
|-------|------|-------|---------|
| K=512 (default) | **43.1** | **26.0** | 33.2 |
| K=64 | 47.0 | 32.3 | 38.7 |
| K=8 | 52.6 | 40.9 | 49.8 |
| K=2 (Taiji-DDN) | 66.5 | 38.4 | 70.6 |
| w/o Split-and-Prune | 55.3 | 31.2 | 34.7 |
| w/o Chain Dropout | 182.3 | 26.5 | 37.4 |
| w/o Learning Res. | 56.2 | 40.2 | 40.2 |
| w/o Leak Choice | 56.0 | 34.3 | **32.2** |

## 4.2 ABLATION STUDY

In Table 2, we demonstrate the impact of different numbers of output nodes ($K$) and various techniques on the network. Interestingly, despite the substantial difference between having and not having the Split-and-Prune technique in the toy example, as shown in fig. 17, the performance in the ablation study without Split-and-Prune is not as poor as one might expect. We hypothesize that this is due to the Chain Dropout forcing all dead nodes to receive gradient guidance,

preventing the network from generating poor results that are unoptimized. A particular case is when $K = 2$, where the representational space of Taiji-DDN is already sufficiently compact. The use of Chain Dropout in this case tends to result in more blurred generated images. Therefore, we did not employ Chain Dropout when $K = 2$.

## 5 CONCLUSION

In this paper, we have introduced Discrete Distribution Networks, a generative model that approximates the distribution of training data using a multitude of discrete sample points. DDN exhibits unique property: more general zero-shot conditional generation. We also proposed the Split-and-Prune optimization algorithm and several effective techniques for training DDN. Additionally, we showcase the efficacy of DDN and its intriguing properties through experiments.

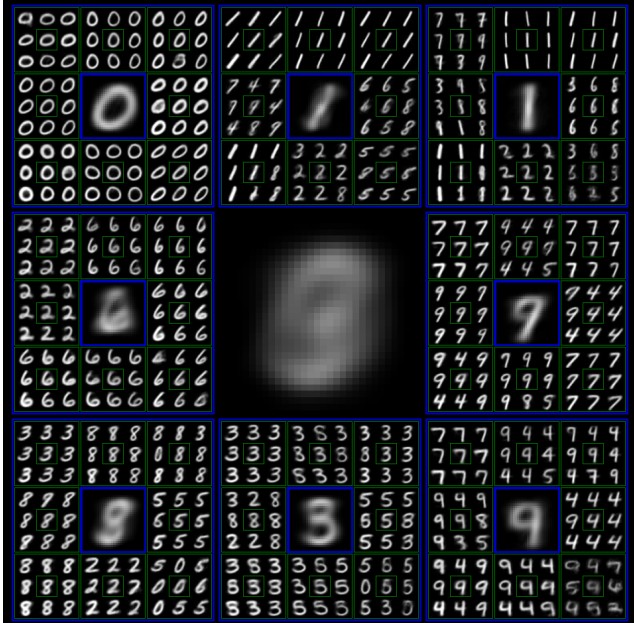

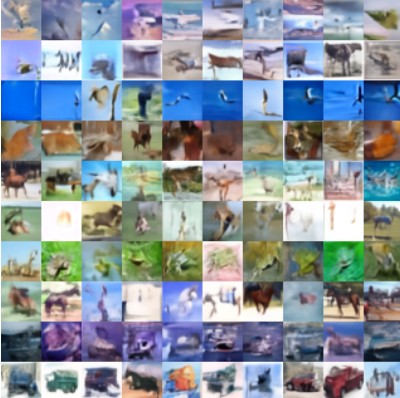

Figure 6: **Hierarchical Generation Visualization of DDN.** We trained a DDN with output level $L = 3$ and output nodes $K = 8$ per level on MNIST dataset, its latent hierarchical structure is visualized as recursive grids. The large image in the middle is the average of all the generated images. Each sample with a colored border represents an intermediate generation image: a blue border indicates generation by the first layer, and a green border by the second layer. The samples within the surrounding grid of each colored-bordered sample are refined versions generated conditionally based on it (enclosed by the same color frontier). The small samples without colored borders are the final generated images. More detailed visualization of $L = 4$ is presented in the appendix fig. 19.

Figure 7: **Zero-Shot Conditional Generation** on CIFAR-10 Guided by a pretrained classification model without gradient. Each row corresponds to a class in CIFAR-10. Specifically, the first row consists of airplanes, the third row displays birds flying against the blue sky, and the last row presents trucks. Our model successfully generates reasonable images for each class without any conditional training, demonstrating the powerful zero-shot generation capability of our DDN.

## 6 ACKNOWLEDGEMENTS

Special thanks to Haikun Zhang for her comprehensive support, including but not limited to paper formatting, CIFAR classification experiments. Thanks to Yihan Xie for her significant assistance in illustrations and CLIP experiments. Gratitude is extended to Jiajun Liang and Haoqiang Fan for helpful discussions. Thanks to Xiangyu Zhang for hosting the Generative Model Special Interest

Group. Appreciation is also given to Hung-yi Lee for developing and offering his engaging machine learning course.

The work was supported by National Science and Technology Major Project of China (2023ZD0121300).

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

## A  EXPERIMENTAL RESULTS ON GENERATIVE PERFORMANCE

We present additional experiments to verify the generative and reconstructive capabilities of Discrete Distribution Networks (DDN).

**Conditional DDN for Image-to-Image Tasks.** In the domain of image-to-image tasks, we extend our Discrete Distribution Networks (DDN) to a conditional setting, resulting in the conditional DDN. The image condition is fed directly into the network during training at the beginning of each stage. The condition serves as an informative guide, significantly reducing the task's generative space and, consequently, mitigating the modeling task's complexity. Through this conditional design, the DDN can leverage the abundant information contained in the conditions to generate more accurate and detailed images as shown in fig. 8.

**Verify whether DDN can generate new images.** As depicted in fig. 9, we compare the images that are closest in the training dataset, FFHQ, to those generated by our DDNs. It suggests that our DDNs can synthesize new images that, while not present in the original dataset, still conform to its target distribution.

**Demonstration of Generation and Reconstruction Quality.** Figures 12 to 14 illustrate the results of generation and reconstruction in various ablation experiments using the DDN model.

**Efficacy of Split-and-Prune and Chain Dropout.** A series of experiments were conducted to separately investigate the effectiveness of the Split-and-Prune and Chain Dropout methods. To isolate the impacts of these two algorithms, we simplified the experimental conditions as much as possible, using the MNIST dataset as a base, setting $K = 8$ and $L = 10$, and disabling Learning Residual. The generated image quality under three different settings is displayed in fig. 10. The results demonstrated that the Split-and-Prune method is indispensable, leading to significant improvements in the quality of generated images. Meanwhile, the Chain Dropout method was found to alleviate the poor results observed when the Split-and-Prune method was not implemented.

**Implementation Details of Split-and-Prune.** In algorithm 1, the Split and Prune operations are executed simultaneously. During implementation, the newly split nodes occupy the positions of the pruned nodes in the tensors, ensuring that the total number of network parameters remains constant and the GPU memory usage is fixed. It is important to note that the algorithm must not only operate

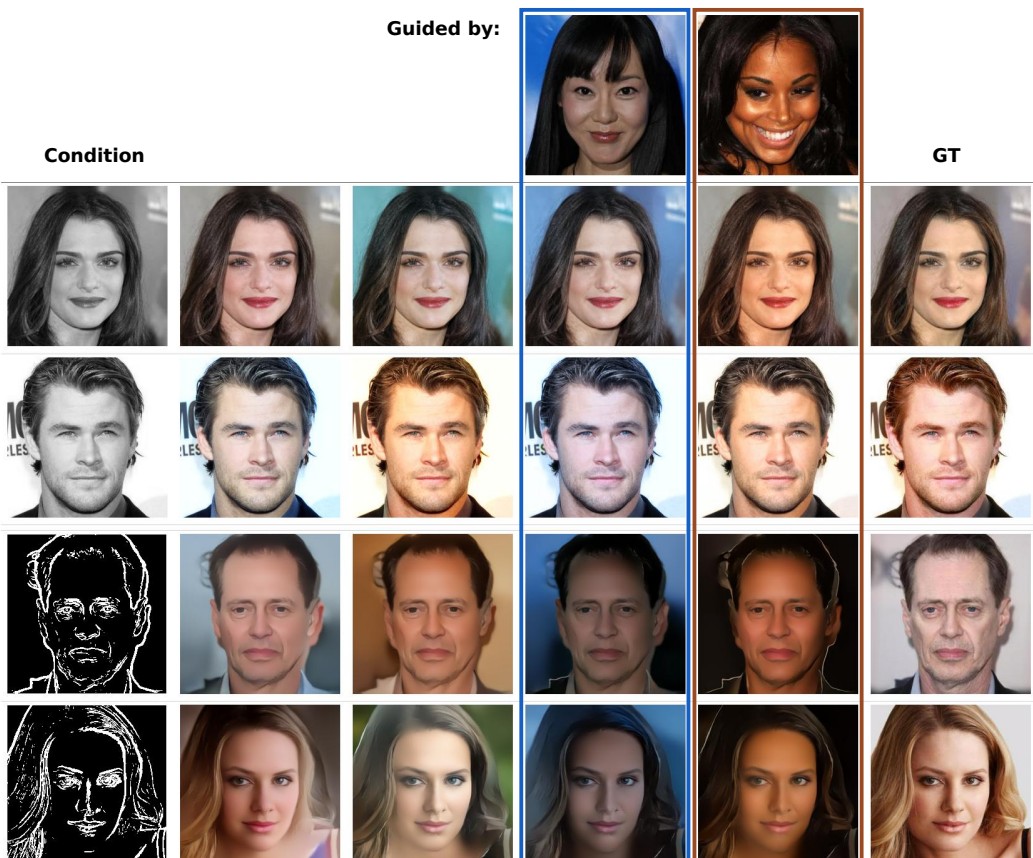

Figure 8: **Conditional DDN performing coloring and edge-to-RGB tasks.** Benefiting from the reduction of the generative space by the condition, DDN is capable of generating high-quality images of $256 \times 256$ resolution. Columns 4 and 5 display combination of conditional generation and ZSCG, the generated results under the guidance of other images, where the produced image strives to adhere to the style of the guided image as closely as possible while ensuring compliance with the condition.

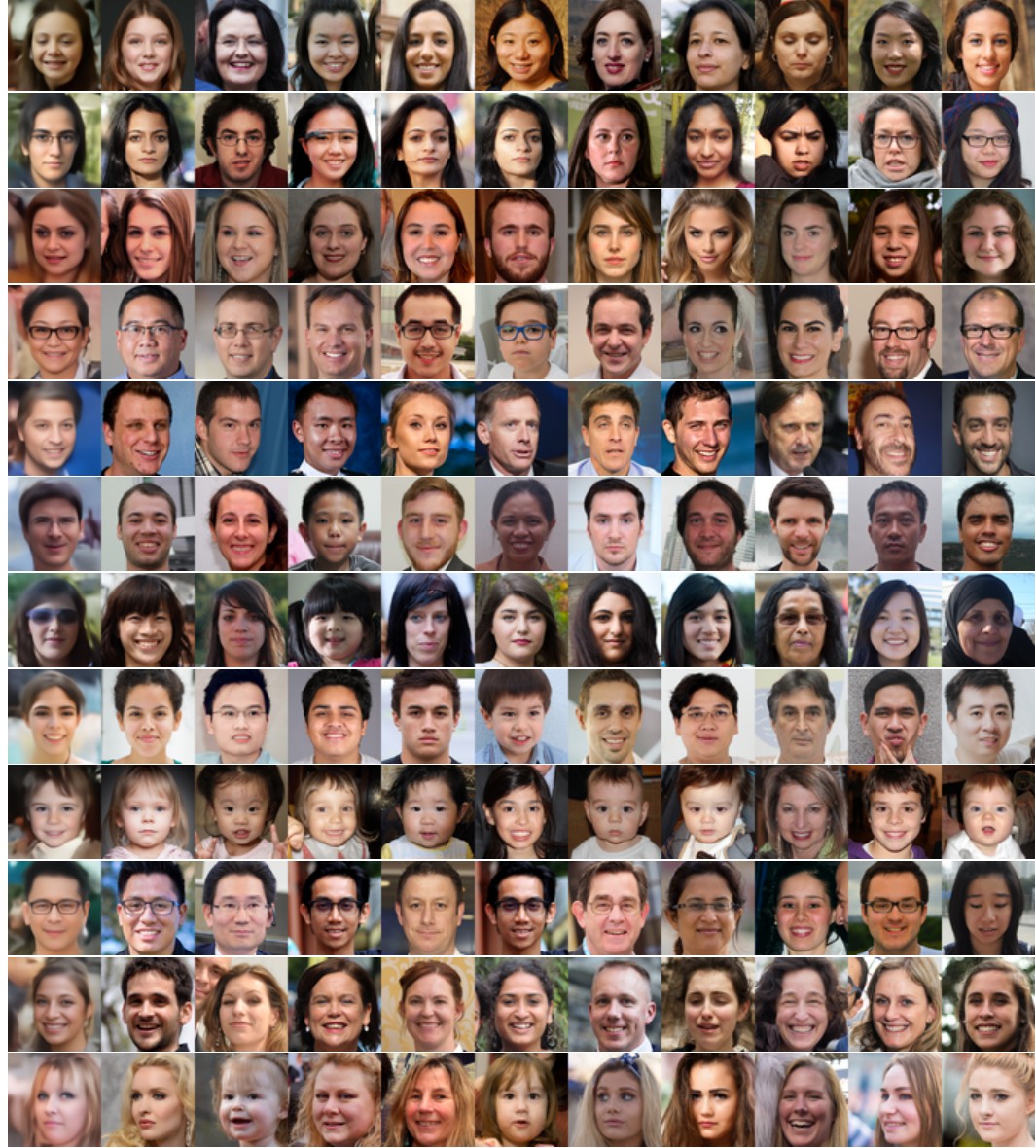

Figure 9: Nearest neighbors of the model trained on FFHQ. The leftmost column presents images generated by the DDN. Starting from the second column, we display the images from FFHQ that are most similar to the generated images, as measured by LPIPS Zhang et al. (2018).

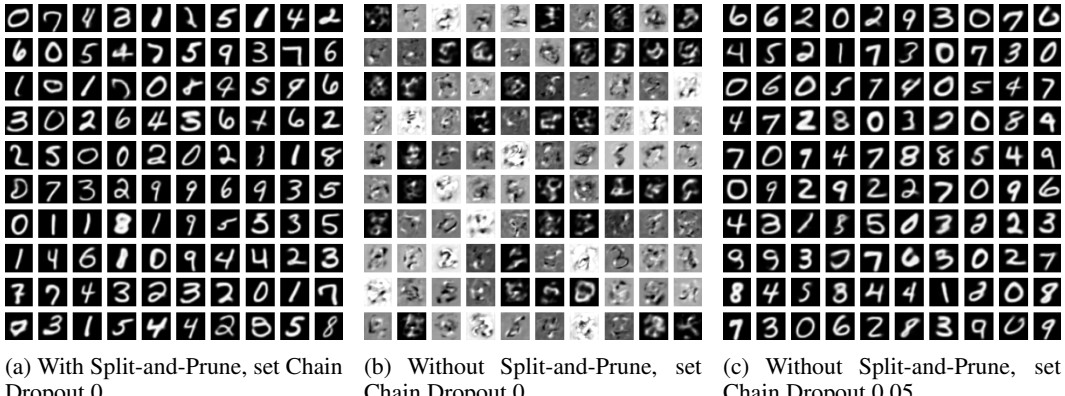

(a) With Split-and-Prune, set Chain Dropout 0

(b) Without Split-and-Prune, set Chain Dropout 0

(c) Without Split-and-Prune, set Chain Dropout 0.05

Figure 10: Efficacy of Split-and-Prune and Chain Dropout on MNIST.

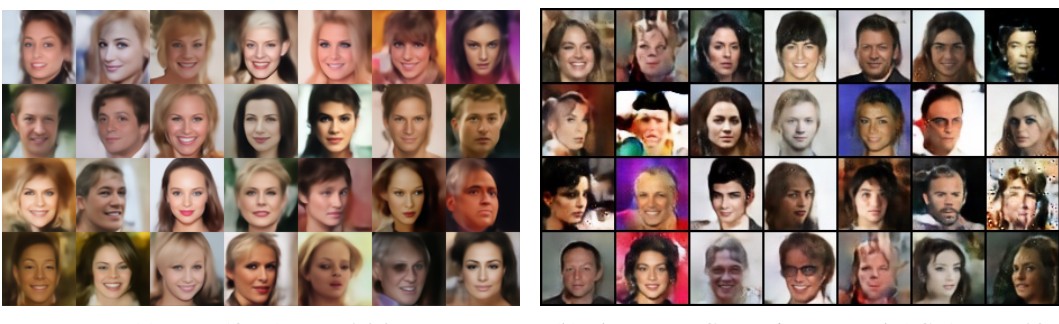

(a) DDN (Ours), FID=35.4

(b) Idempotent Generative Network (IGN), FID=39

Figure 11: **Comparison of randomly generated images on CelebA-HQ 64x64.** DDN produces images with clearer details and fewer artifacts compared to IGN Shocher et al. (2024).

on the output parameters but also apply the same operations to the corresponding state variables in the optimizer. This ensures consistency between the parameters being optimized and their associated state information throughout the training process.

**Comparison with other novel generative models.** We conduct a qualitative comparison with a recent work, the Idempotent Generative Network (IGN) Shocher et al. (2024), accepted by ICLR 2024. Since IGN was only experimented on the CelebA dataset and did not release its code, our comparison is limited on the CelebA dataset. As depicted in fig. 11, our DDN demonstrate better generation capabilities over IGN.

# B FURTHER DEMONSTRATIONS ON ZERO-SHOT CONDITIONAL GENERATION

In this section, we present additional experimental results on Zero-Shot Conditional Generation (ZSCG).

**Utilizing CLIP as Conditioning Guidance.** As illustrated in fig. 15, we use CLIP Radford et al. (2021) along with the corresponding prompts as conditions. We use a DDN model, which is exclusively trained on the FFHQ dataset, to yield Zero-Shot Conditional Generation (ZSCG) results. The results indicate that, DDN can generate corresponding images under the guidance of CLIP without the necessity for gradient computations.

**ZSCG with Multiple Conditions.** In fig. 16, we illustrate the operation of ZSCG under the combined action of two Guided Samplers: Inpainting and CLIP. Each sampler operates under its own specific condition. The inpainting sampler utilizes a mask to cover the areas where the CLIP prompt

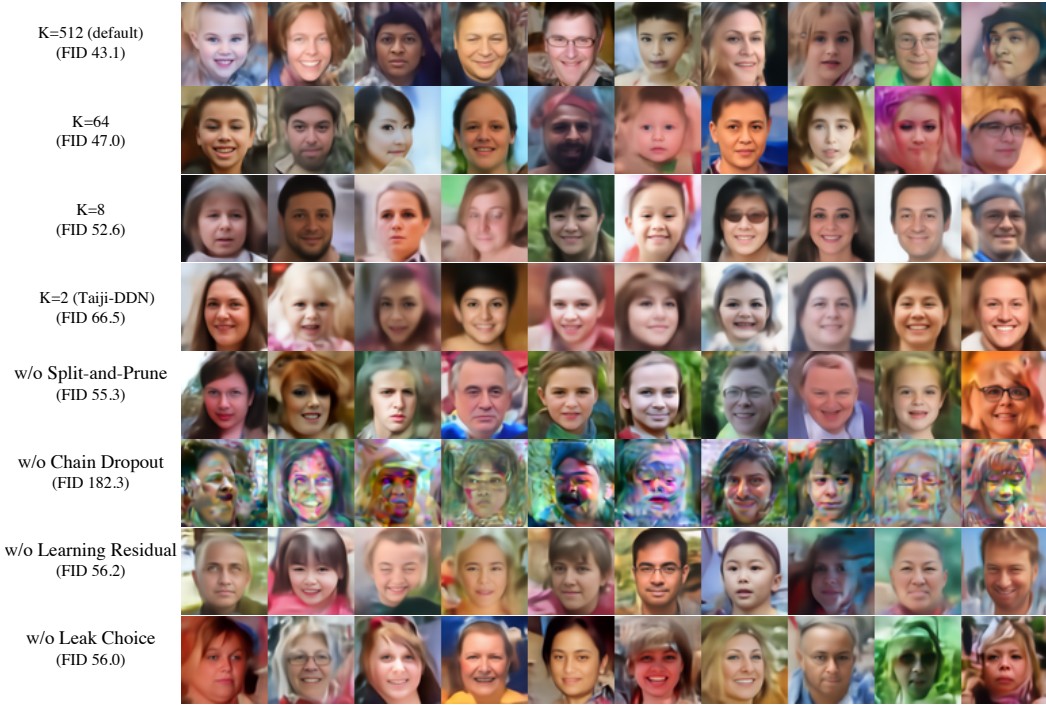

Figure 12: Illustration of the random sample generation effects as part of the ablation study on our DDNs model.

| Nodes/level | Level | Representation space | Validation accuracy↑ | | | |
|---|---|---|---|---|---|---|
| K | L | $K^L$ | 128 | 1024 | 10k | 50k |
| 2 | 10 | 1024 | 65.9 | 77.3 | 85.8 | 86.9 |
| 8 | 3 | 512 | **69.0** | **81.1** | **87.6** | 88.0 |
| 8 | 5 | $3.3 \times 10^4$ | 67.5 | 79.1 | 87.5 | **90.5** |
| 8 | 10 | $1.1 \times 10^9$ | 58.6 | 75.3 | 84.9 | 89.0 |
| 64 | 10 | $1.1 \times 10^{18}$ | 52.5 | 70.4 | 80.9 | 86.3 |

Table 3: **Fine-tuning DDN latent as decision tree on MNIST.** Constructing a decision tree based on the latent variables from the DDN and fine-tuning it on MNIST trainning set. We report the validation set accuracy of the decision tree after majority voting for class prediction with varying number of training samples: 128, 1,024, 10,000, and 50,000 (the full training set).

acts. Specifically, "wearing sunglasses" masks the eyes, "wearing a hat" masks the upper half of the face, and "happy person" masks the lower half.

For each Discrete Distribution Layer (DDL), both samplers assign a rank to every generated image, corresponding to the degree of match to their respective conditions– the better the match, the higher the rank. We apply a weight to each sampler, which represents their influence on the final assigned ranking. In this case, both samplers have a weight of 0.5. To promote diversity in the generated samples, we randomly select one image from the top two ranked generated images, serving as the output for that DDL layer.

## C LATENT ANALYSIS

**Semantic Performance of Latents.** We explored the semantic capabilities of DDN latents through a classification experiment on the MNIST dataset. Given the inherent tree structure of DDN's latents, we employed a decision tree classification method, using fine-tuning data to assign class votes to nodes in the tree. For unassigned nodes, their class is inherited from the closest ancestor node with an assigned class. We fine-tuned DDN's latent decision tree using various numbers of labeled training set data, and the results on the test set are shown in table 3. All experiments in this table were

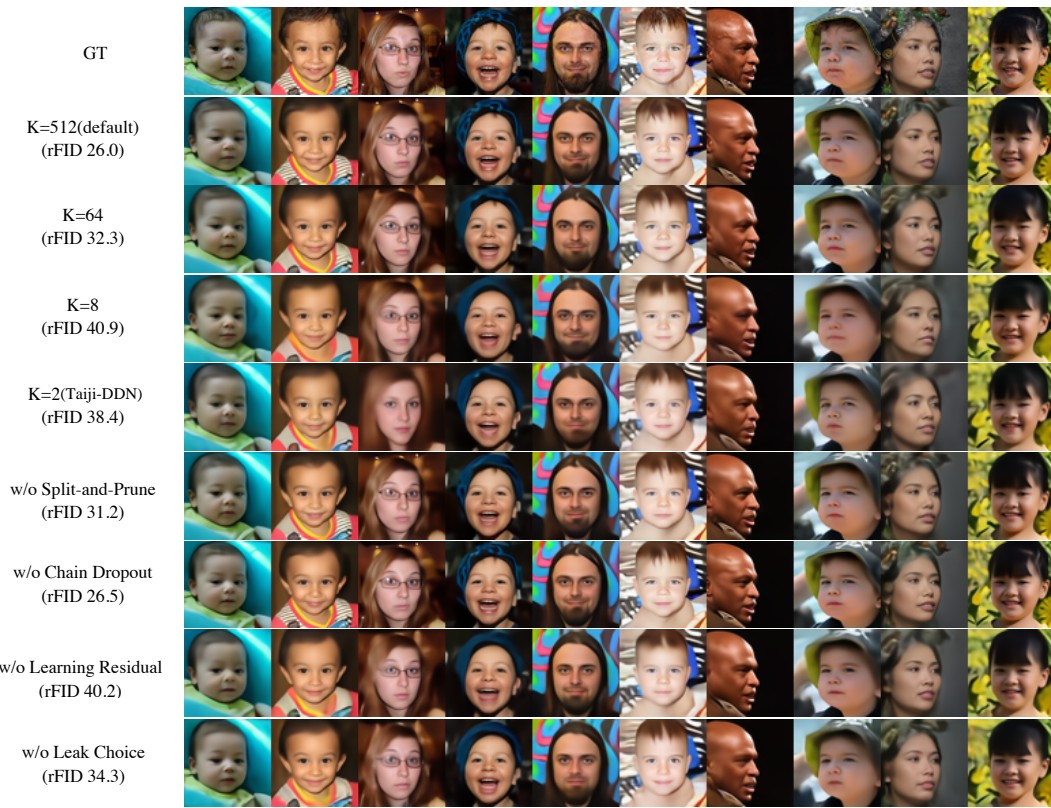

Figure 13: Demonstration of the reconstruction capability of our ablation study model on FFHQ-64x64, which can be interpreted as the model's fitting ability on the training set.

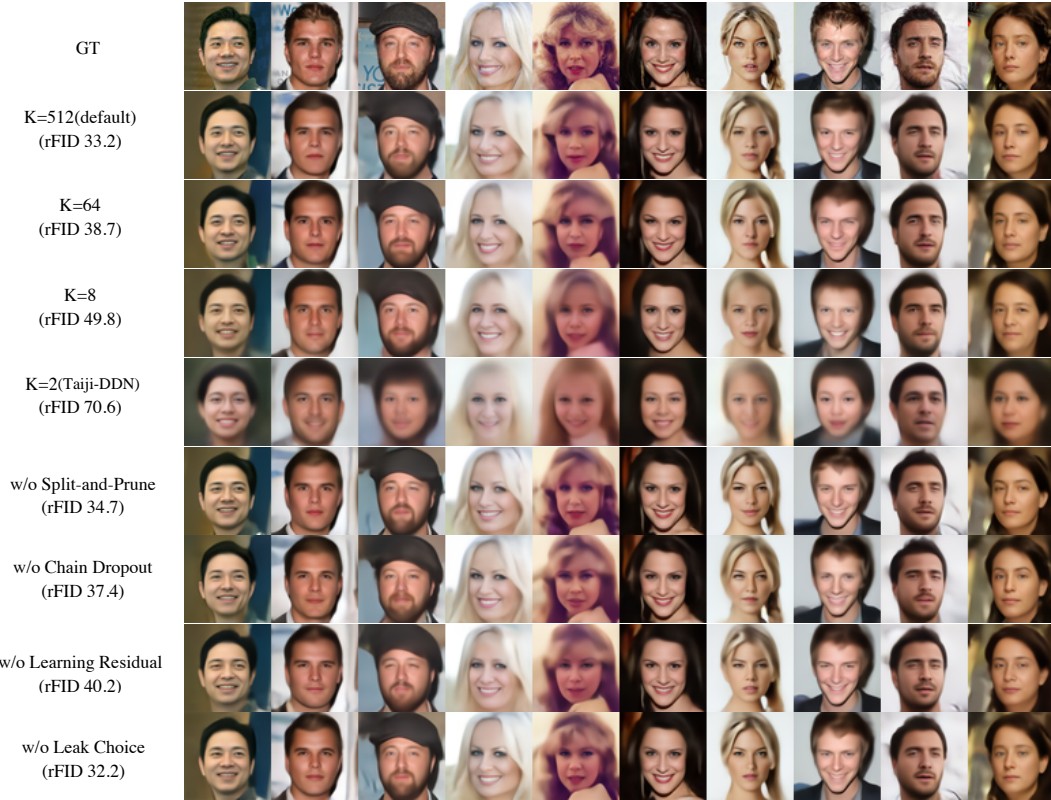

Figure 14: Demonstration of the reconstruction capability of our ablation study model on CelebA-64x64, which can be interpreted as the model's generalization ability on the test set.

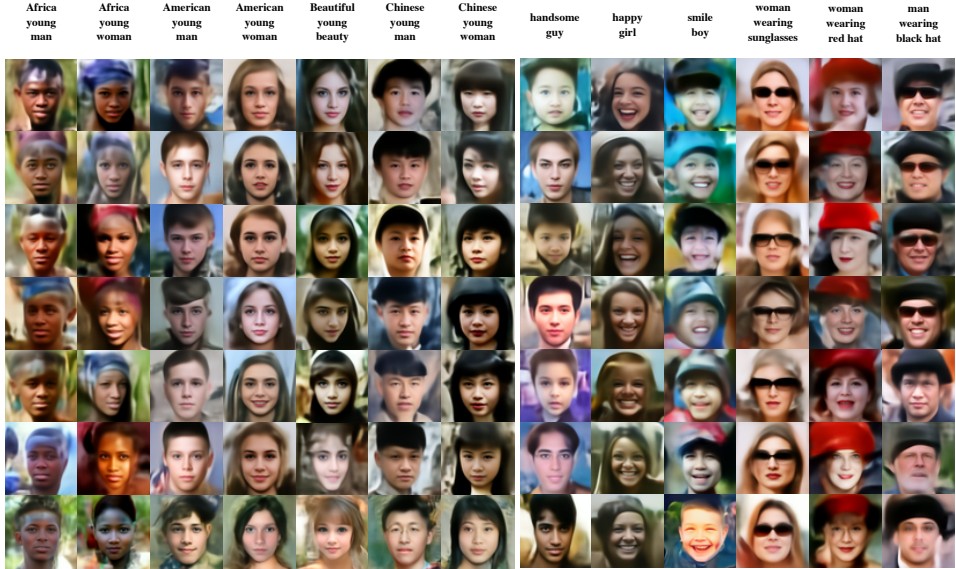

Figure 15: **Zero-Shot Conditional Generation guided by CLIP.** The text at the top is the guide text for that column.

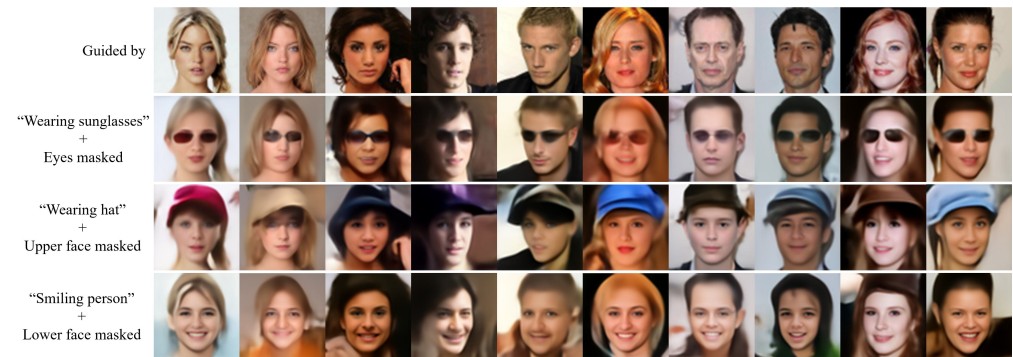

Figure 16: **Zero-Shot Conditional Generation under the Influence of Multiple Conditions.** The DDN balances the steering forces of CLIP and Inpainting according to their associated weights.

conducted with Recurrence Iteration Paradigm's UNet, which has approximately 407k parameters. These experiments substantiate that DDN's latents encompass meaningful semantic information.

**More Comprehensive Latent Visualization.** fig. 19 demonstrates a more comprehensive distribution of samples that correspond to the latent variables.

## D  MORE DETAILED EXPERIMENTAL EXPLANATION

In fig. 20, we have expanded the "Illustration of the principle behind the Split-and-Prune operation" by providing a schematic when $K$ increases to 15. This demonstrates that a larger sample space yields an approximation closer to the target distribution.

In the caption of fig. 17, we have detailed the experimental parameters for "Toy examples for two-dimensional data generation". Additionally, we will explain why the KL divergence in our model is lower than that found in the Real Samples.

## E  LIMITATIONS AND FUTURE WORK

There are some key limitations of Discrete Distribution Networks (DDNs):

- **The $K^L$ output space is insufficiently large to represent complex distributions.**: We are currently developing a new approach to enhance the efficiency of high-dimensional data representation by expanding the output space from $K^L$. This involves dividing an image into $N$ patches, independently selecting the optimal patch from $K$ candidates for each patch. These selected patches are then merged into a complete image, which serves as the output of the current DDL layer and is input into the next layer. Consequently, this increases the output space to $K^{N \cdot L}$.

- **The Split-and-Prune algorithm continuously discards trained parameters.**: Discarding parameters that have undergone extensive training is not a wise choice, particularly when scaling up the model. The goal of the Split-and-Prune algorithm is to balance the frequency at which each node is sampled during training, similar to maintaining load balance among experts in Mixture-of-Experts (MoE) models Liu et al. (2024). A potential solution to address this issue is the Loss-Free Balancing method Wang et al. (2024). After computing the L2 distance between each node's output and the ground truth, Loss-Free Balancing first applies a node-wise bias to these distances. By dynamically updating the bias of each node based on its recent load, this method ensures that nodes with lower sampling frequencies receive a lower bias, thereby increasing their chances of being sampled. Consequently, Loss-Free Balancing helps maintain a balanced distribution of node sampling without discarding parameters.

- **Loss of high-frequency signals**: The high level of data compression and the use of pixel L2 loss during optimization may result in the loss of high-frequency signals, causing the

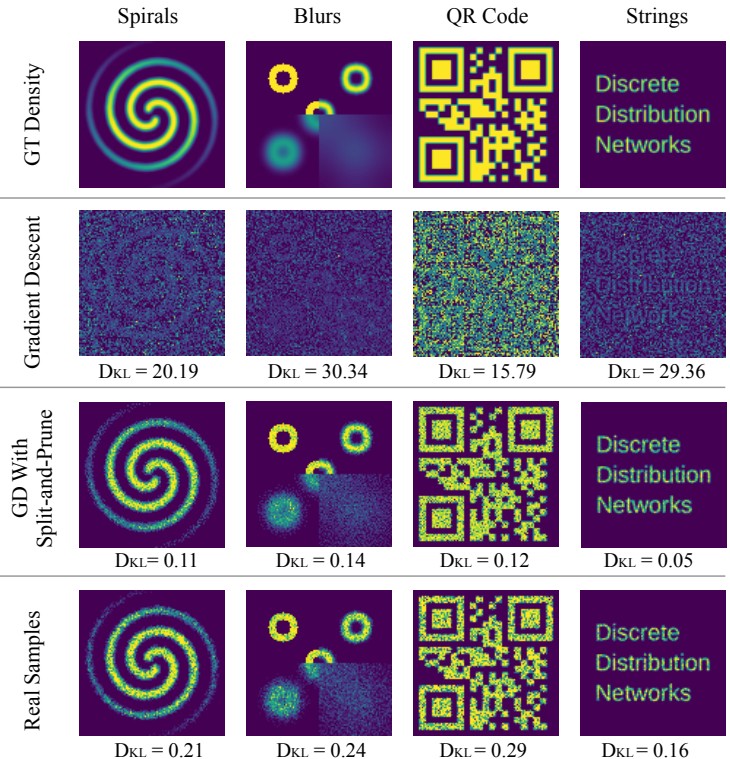

Figure 17: **Toy examples for two-dimensional data generation.** The numerical values at the bottom of each figure represent the Kullback-Leibler divergence, $D_{KL}$. Due to phenomena such as "dead nodes" and "density shift", applying Gradient Descent alone fails to properly fit the Ground Truth (GT) density. By employing the Split-and-Prune strategy, the density map looks the same as Real Samples. In the experiment, we use $K = 10,000$ discrete nodes to emulate the probability distribution of the GT density. Each node encompasses two parameters, $x$ and $y$, initialized from a uniform distribution. Each experiment consists of $10 \times K = 100,000$ iterations, wherein each iteration, an L2 loss is calculated based only on the node closest to the GT. To compute the KL divergence, the GT density map is converted into a discrete distribution with bins of size $100 \times 100$, which is then used to calculate the $D_{KL}$ against the discrete distribution represented by these nodes. Notably, the KL divergence of Split-and-Prune is even lower than that of the Real Samples. This is because our algorithm has been exposed to $10 \times K$ samples of the GT distribution, thus it better reflects the GT distribution compared to the Real Samples, which are drawn only $K$ times from the GT distribution.

太极生两仪，两仪生四象，
四象生八卦，八卦生万物。

《易经》

From the Taiji generates the Two Poles;
From the Two Poles generate the Four Symbols;
From the Four Symbols generate the Eight Trigrams;
And from the Eight Trigrams generate the Myriad Things.

*I Ching: The Book of Changes*

(a) Famous quote and its translation about Taiji

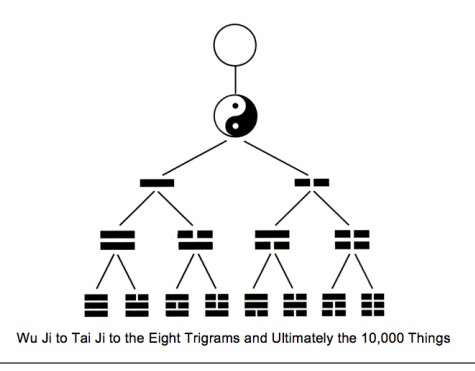

(b) Tree structure of Taiji

Figure 18: **The Taiji-DDN exhibits a surprising similarity to the ancient Chinese philosophy of Taiji.** Records of Taiji can be traced back to the I Ching (Book of Changes) from the late 9th century BC, often described by the quote on the left (a) that explains the universe's generation and transformation. This description coincidentally also summarizes the generation process and the transformations in the generative space of Taiji-DDN. Moreover, the diagram (b) from the book Tom (2013) bears a closely resemblance to the tree structure of DDN's latent fig. 1b. Therefore, we have named the DDN with $K = 2$ as Taiji-DDN.

images to appear blurred. A potential improvement could be learning from VQ-GAN Esser et al. (2021) and incorporating adversarial loss Creswell et al. (2018) to enhance the modeling of high-frequency signals.

- **Computational burden of zero-shot conditional generation**: ZSCG requires $L \times K$ forward passes through the guided model, where $L$ is the number of layers and $K$ is the number of possible outputs per layer. When the guided model itself is computationally expensive, this results in significant computational overhead and prolonged generation time. However, since the discrimination process is parallelizable across the $K$-dimensional output space, batching techniques can be employed to mitigate latency. In addition, it is not necessary for every layer to be guided by the condition signal, further research will be conducted to reduce the number of guided model calls during the ZSCG process.

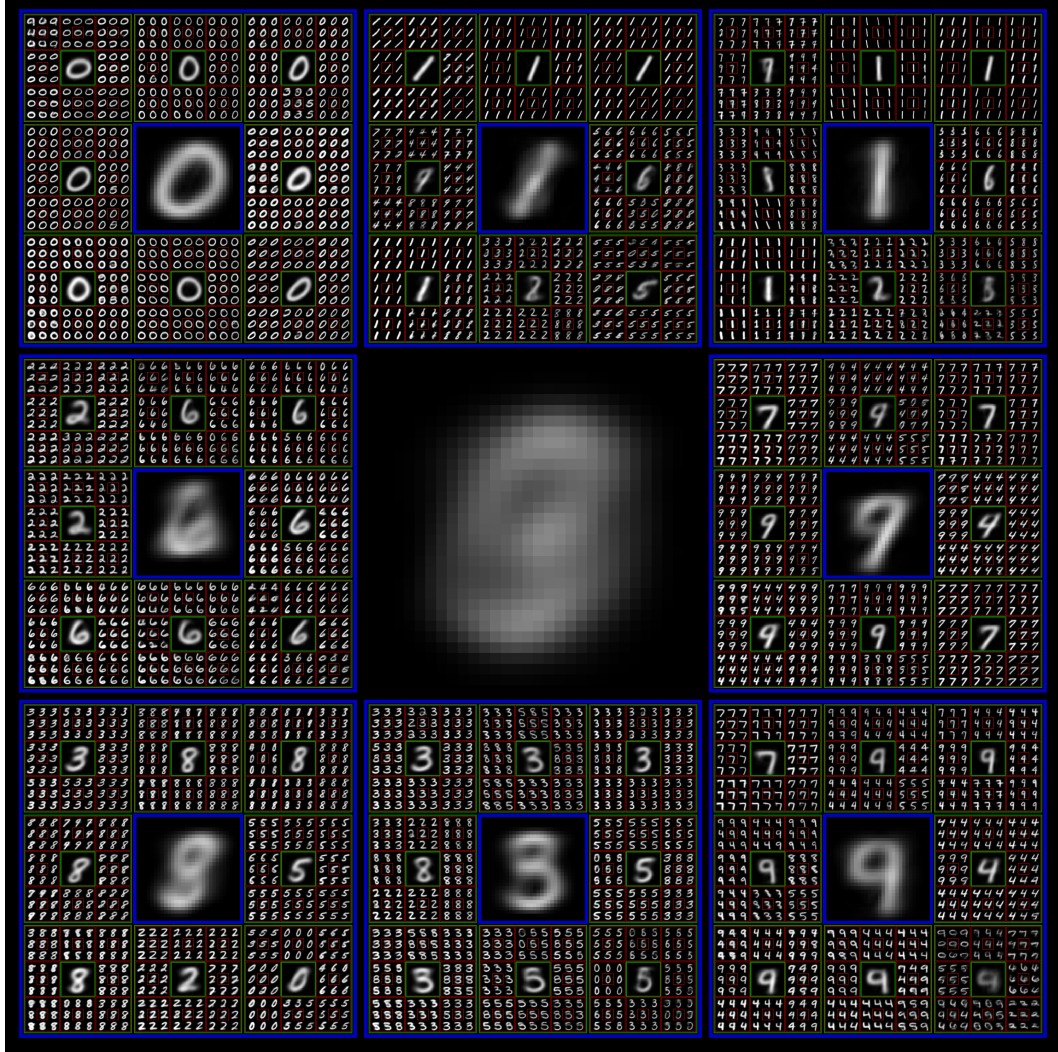

Figure 19: **Hierarchical Generation Visualization of DDN with** $L = 4$. We trained a DDN with output level $L = 4$ and output nodes $K = 8$ per level on MNIST dataset, its latent hierarchical structure is visualized as recursive grids. Each sample with a colored border represents an intermediate generation product. The samples within the surrounding grid of each colored-bordered sample are refined versions generated conditionally based on it (enclosed by the same color frontier). The small samples without colored borders are the final generated images. The larger the image, the earlier it is in the generation process, implying a coarse version. The large image in the middle is the average of all the generated images. The samples with blue borders represent the 8 outputs of the first level, while those with green borders represent the $8^2 = 64$ outputs of the second level. It can be observed that images within the same grid display higher similarity, due to their shared "ancestors". Best view in color and zoom in.

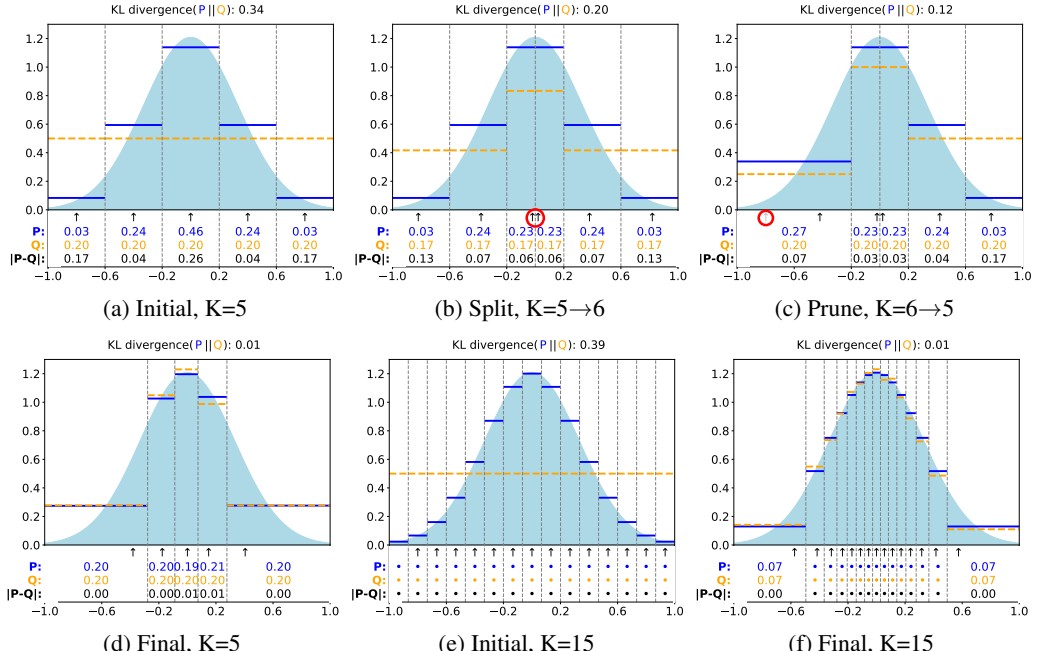

Figure 20: **Illustration of the principle behind the Split-and-Prune operation.** For example in (a), the light blue bell-shaped curve represents a one-dimensional target distribution. The 5 "↑" under the x-axis are the initial values from a uniform distribution of 5 output nodes, which divide the entire space into 5 parts using midpoints between adjacent nodes as boundaries (i.e., vertical gray dashed lines). Each part corresponds to the range represented by this output node on the continuous space $x$. Below each node are three values: $P$ stands for the relative frequency of the ground truth falling within this node's range during training; $Q$ refers to the probability mass of this sample (node) in the discrete distribution output by the model during the generation phase, which is generally equal for each sample, i.e., $1/K$. The bottom-most value denotes the difference between $P$ and $Q$. Colorful horizontal line segments represent the average probability density of $P$, $Q$ within corresponding intervals. In (b), the Split operation selects the node with the highest $P$ (circled in red). In (c), the Prune operation selects the node with the smallest $P$ (circled in red). In (d), through the combined effects of loss and Split-and-Prune operations, the distribution of output nodes moves towards final optimization. From the observed results, the KL divergence ($KL(P||Q)$) consistently decreases as the operation progresses, and the yellow line increasingly approximates the light blue target distribution. Finally, we show the distributions of the initial and final stages when the number of output nodes $K = 15$ in (e) and (f). Due to the increased representational space, the generated probability distribution $Q$, represented by the yellow line segment in (f), is closer to the light blue target distribution than in the case of $K = 5$.