# OpenReview forum: "Discrete Distribution Networks"
_ICLR.cc/2025/Conference — ICLR 2025 Poster_

### Official Review · Reviewer_r4YK · 2024-10-29

**Soundness:** 4
**Presentation:** 4
**Contribution:** 4
**Rating:** 8
**Confidence:** 4

**Summary:**

The authors propose a new generative model, based on generating K examples at each layer and choosing the one closest to a ground-truth instance. This formulates a discrete hierarchical distribution. At inference, taking a random node at each layer provides a random sample. Both conditional and unconditional generation are demonstrated.

**Strengths:**

1. I find the method novel and elegant.
2. The ability to produce visualization of the hierarchical generation (figs 8, 18) is a very enlightening feature.
3. The authors propose some practical techniques to deal with this non-differentiable sampling. The proposed "split-and-prune" trick is clever and elegant.
4. The novelty is very strong, and this should not be overlooked. This is a whole new method, very different from any of the existing generative models.

**Weaknesses:**

1. Theoretical analysis is missing. No mathematical derivation that shows why the distribution of generated images should converge to the real data distribution.
2. Eventually, if I understand correctly, for unconditional generation, the model has a finite number of specific examples it can produce K^L. Even if this number is big, this suggests that the model essentially is a compressed way of storing all its possible results in a tree based database. This might make a problem to scale up. Since the demonstrated data relatively has few dimensions, this suggests that actually holding the entire dataset might require more or close storage to the model itself. A quick calculation to demonstrate: MNIST has 70k images (train+test) of 28x28, this is 54880000 integers: ~55MB. EDM smallest model has 62M parameters, even if we take these as 1 byte each it is heavier.
3. I am adding the weakness of low quantitative results and lack of demonstration of higher scales. To be clear, I think it is legitimate for novel methods but it is still a weakness. (i.e. if this model would have produced SotA generation results it would have been rated higher).

**Questions:**

While has some notable weaknesses, I think this is a very good paper that can open a door to new directions in generative modeling. It is important for me to point to the courage of trying new refreshing approaches (as opposed to building upon current trends). Such papers should always be judged under the understanding that for existing methods, a lot of engineering has taken place and should be thought of as the first GAN paper or the first Diffusion models paper (Sohl-Dickstein 2015).

---

> ### Author Response · Authors · 2024-11-20
> **Official Comment by Authors**
>
> Thank you for your inspiring feedback and appreciation.
>
> **“Theoretical analysis is missing. No mathematical derivation that shows why the distribution of generated images should converge to the real data distribution.”**:
> We primarily demonstrate the effectiveness of the Gradient Descent with Split-and-Prune algorithm through detailed illustrations and 2D toy experiments. We acknowledge the necessity for further research to understand and formalize the underlying mechanisms.
>
> **“Even if this number is big, this suggests that the model essentially is a compressed way of storing all its possible results in a tree based database. This might make a problem to scale up. Since the demonstrated data relatively has few dimensions, this suggests that actually holding the entire dataset might require more or close storage to the model itself. A quick calculation to demonstrate: MNIST has 70k images (train+test) of 28x28, this is 54880000 integers: ~55MB. EDM smallest model has 62M parameters, even if we take these as 1 byte each it is heavier.”**:
> Admittedly, we are not entirely clear about the background and motivation of this example, but we will attempt to address it from several perspectives:
> - DDNs perform lossy compression, meaning they do not store every byte of the dataset precisely. Consequently, when reconstructing from the latent space, the generated images will differ from the originals (measurable by L2 loss). Thus, the notion of "actually holding the entire dataset" may not apply directly; instead, the model's parameter count influences the reconstruction accuracy (reflected in reconstruction loss and rFID).
> - Additionally, latents also contribute information. Although the original ground truth (GT) does not directly input into the network, it influences the DDN through the guided sampling process, transmitting information in latent form. Hence, both network parameters and latents collectively enable the reconstruction of training samples, meaning the dataset's information is carried by both the model parameters and the latents of the entire dataset.
> - Furthermore, we believe that under the same storage space, the intrinsic information content or complexity of different datasets can vary significantly. Most MNIST pixel values are either 0 or 255, and black pixels near the edges are deterministic and thus carry no information, allowing the model to fit MNIST well. However, replacing this with CIFAR-style images of the same pixel count would significantly increase the modeling difficulty. Therefore, solely relying on the uncompressed storage space of the dataset to reflect the fitting difficulty of a model with a certain parameter size can lead to substantial discrepancies.
>
>
> **“I am adding the weakness of low quantitative results and lack of demonstration of higher scales. To be clear, I think it is legitimate for novel methods but it is still a weakness.”**:
> As you noted, achieving state-of-the-art (SoTA) results with a novel method like DDN requires more research and effort. We are currently focusing on enhancing the generation quality of our model for Zero-Shot Conditional Generation (ZSCG) tasks.
>
> We hope this response addresses your concerns. Thank you for considering our work.

---

> > ### Comment · Reviewer_r4YK · 2024-11-23
> > **Thank you for your response**
> >
> > * **Theoretical analysis:** Not sure how demonstrating effectiveness relates to this concern. The authors acknowledge that formalization is missing. The basic mathematical derivation needed is assuming perfect convergence and expressiveness of the model, having data distribution $P(x)$ and show that the generated instances from the model $y=f(z), P(y)\approx P(x)$   ($z$ in this case somehow describes the stochaticness of the sampling layers). Such analysis is the core of the theory of generative models.
> >
> >
> > * **Finite possible generated instances:** Sorry if my previous comment was not clear about this. The point was skepticism about the ability to scale up. I was pointing out that there is a finite number of possible outputs from the model: $K^L$. The purpose of the MNIST example was to demonstrate that holding the entire dataset actually weighs less than the model weights. Of course, the practical use of generative models is not to generate existing examples from the dataset. However analytically, the perfect generative model is the one that produces the dataset exactly. My skepticism about scaling up arises from the fact that the finite number of possible examples may require more and more layers as the dataset grows bigger, making it impractical for the modern gigantic datasets. Since empirically it was not shown that the model is capable of high-scale generation and theoretically it seems limited I remain skeptical about scaling it up.
> >
> >
> > * **General:** My concerns were addressed and I appreciate it, but I was not convinced with any of the responses. However, the rating I granted was fully taking into account these concerns. These concerns are significant and I really hope the authors are able to fix them for this submission or for a future one. However, I meant what I wrote in the summary of my original review-  Many high rated papers would have been done by someone else if their authors never published them or were rejected. However, if this paper is not published, it is not likely that anyone would come up with this approach. This is real publication value. I am reminding again the original diffusion paper from 2015 (Sohl-Dickstein) that was almost not noticed for 5 years. Had it not been published, would we have had the amazing generative models we have today?
> >
> > I hope my concerns are clearer now. My suggestion to the authors: it is unlikely, at this point, that I will upgrade my already very good rating. I also have no intention of downgrading it. I am supportive of this paper. Addressing successfully to my concerns may strengthen the case for the final decision, but the best use of your time is addressing other reviews that are less favorable.
> > Thanks!

---

> > > ### Author Response · Authors · 2024-11-28
> > > **Thanks and share new approach to expand the latent space**
> > >
> > > We are currently developing a new approach to expand the latent space from $K^L$. This approach involves dividing an image into $N$ patches, independently selecting the optimal patch from $K$ candidates for each patch, and then combining these selected patches into an image. This image serves as the output of the current DDL layer and is input into the next layer, thereby expanding the latent space to $K^{N \cdot L}$.
> > >
> > > Finally, thank you again for your detailed guidance and active discussions.

---

### Official Review · Reviewer_1dyp · 2024-11-01

**Soundness:** 3
**Presentation:** 3
**Contribution:** 3
**Rating:** 8
**Confidence:** 4

**Summary:**

This work proposes Discrete Distribution Networks (DDN) a new class of image generative models. Discrete Distribution Layers within a DDN output multiple images at a time, and one image is sampled and concatenated as a feature to input to the next layer in an autoregressive fashion. During training, the loss is a reconstruction loss of the output sample of each DDL that best matches with the input image. Several tricks are provided to help optimization. During sampling, external signal can be provided without using gradients in order to condition generation. Qualitative and quantitative results are provided for unconditional CIFAR-10, FFHQ, and Celeb-A generation in addition to qualitative results for MNIST.

**Strengths:**

* The idea of sampling an image and autoregressively feeding it back in through each layer in a forward pass is novel to the best of my knowledge.
* The ability to condition generation on external signals without gradients is a useful property.
* DDN is showcased for a variety of applications such as inpainting, colorization, denoising, super-resolution, CLIP-guided editing.
* A variety of tricks are provided to help training, such as the Split-and-Prune algorithm, chain dropout, residual learning, leak choice.

**Weaknesses:**

* I understand that this paper serves as a proof-of-concept for a new type of generative model, but the model is validated on small scale datasets (e.g., CIFAR-10, MNIST, CelebA, FFHQ-64x64). Furthermore, it does not compare against more modern approaches such as diffusion models, and quantitatively lags behind DC-GAN. However, I do not think it is fair to hold this novel method to the same standard as more matured, modern approaches.
* The method relies on generating multiple samples in parallel and requires multiple layers for further refinement. This seems quite expensive and hard to scale to higher resolutions.
* The placement of tables and figures reduces the quality of presentation. For instance, Figure 9 has a large white space above it. Table 2 is presented on page 8 even though ablations are not addressed in text until page 10. Figure 5 showcases an experiment which is not addressed in the main text of the paper. So it would be more fitting in the appendix.

**Questions:**

* Can DDN scale to larger and higher resolution datasets, such as FFHQ 256x256, or ImageNet 256x256?
* What are the parameter counts of the baselines compared (DC-GAN, IGEBM, VAE, etc.)? DDN benefits from more layers since it can refine more, so the baselines should have similar parameter counts.
* How does a single shot DDN compare against a default DDN allowing for the same number of refinements?
*  Generating multiple samples in parallel for sampling is expensive. What are the memory requirements compared to the baselines in the paper?

---

> ### Author Response · Authors · 2024-11-20
> **Official Comment by Authors - (1/2)**
>
> Thank you for the thoughtful consideration of our paper and the constructive feedback.
>
>
> **1 “model is validated on small scale datasets,  it does not compare against more modern approaches”**:
> As a novel method, achieving state-of-the-art (SoTA) performance with DDN requires further research and effort. We are currently focusing on enhancing the generation quality of our model for Zero-Shot Conditional Generation (ZSCG) tasks.
>
> **2 “The method relies on generating multiple samples in parallel and requires multiple layers for further refinement. This seems quite expensive and hard to scale to higher resolutions.”**:
> - Due to the hierarchical conditional generation approach, similar to autoregressive(AR) models, DDN faces same challenges as AR models in generating high-quality, high-resolution images, which significantly increases computational cost.
> - We are currently developing a new approach to enhance the efficiency of high-dimensional data representation by expanding the latent space from $K^L$. This involves dividing an image into $N$ patches, independently selecting the optimal patch from $K$ candidates for each patch, and then merging these selected patches into an image. This image serves as the output of the current DDL layer and is input into the next layer, thereby increasing the latent space to $K^{N \cdot L}$.
> - Another potential solution is to draw inspiration from Latent Diffusion and VQ-VAE, enabling DDN to perform generative modeling in a lower-complexity latent space.
>
> **3 “The placement of tables and figures reduces the quality of presentation.”**:
> We have revised the manuscript to improve its clarity and flow. Specifically, we have moved Figure 5 (2-d toy example) to the Appendix to streamline the main text. Additionally, Table 2 has been relocated to be adjacent to Section 4.2, the Ablation Study, to enhance the logical coherence and accessibility of relevant data.
>
>
> **4 “Can DDN scale to larger and higher resolution datasets, such as FFHQ 256x256, or ImageNet 256x256?”**:
> - In Appendix Figure 9, we demonstrate conditional generation on FFHQ 256x256, where additional information helps reduce the generative space, allowing DDN to produce clear 256-resolution images.
> - For unconditional FFHQ 256x256, the current version of DDN can generate recognizable faces but with blurred contours and lacking high-frequency details.
> - The current version of DDN struggles to generate coherent ImageNet 256x256 images, primarily due to the small model size (74M parameters). Scaling up the model parameters is necessary. For reference, Image GPT (https://openai.com/index/image-gpt/) uses a 1.4B parameters model to generate 96x96 ImageNet images.
>
>
> **5 “What are the parameter counts of the baselines compared?”**:
> We have supplemented the parameter counts based on the survey paper on generative models (https://arxiv.org/abs/2103.04922):
> | Method | Type | FID↓ | param(P) |
> | --- | --- | --- | --- |
> | DCGAN | GAN | 37.1 | P < 10M |
> | IGEBM | EBM | 38.0 | P < 10M |
> | VAE | VAE | 106.7 | P < 10M |
> | Gated PixelCNN | AR | 65.9 | P > 120M |
> | GLOW | Flow | 46.0 | 60M < P < 120M |
> | DDN(ours) | DDN | 52.0 | P=74M |

---

> ### Author Response · Authors · 2024-11-20
> **Official Comment by Authors - (2/2)**
>
> **6 “How does a single shot DDN compare against a default DDN allowing for the same number of refinements?”**:
> - There might be some misunderstanding here. The single-shot paradigm is our default paradigm, leveraging its coarse-to-fine nature to enhance computational efficiency. Only the MNIST experiment used the recurrence iteration paradigm of DDN to demonstrate its effectiveness. To avoid confusion, we have revised the paper to emphasize that single-shot is the default paradigm for most experiments.
> - In comparing single-shot paradigm with the recurrence iteration paradigm, given the same parameter count and number of refinements, recurrence paradigm requires L times more computation since each parameter is processed L times to obtain the final result. Moreover, the recurrence paradigm cannot effectively utilize the coarse-to-fine nature, resulting in lower computational efficiency. Thus, the single-shot paradigm is a more favorable choice.
>
> **7 “Generating multiple samples in parallel for sampling is expensive. What are the memory requirements compared to the baselines in the paper?”**:
> - Our memory requirements are slightly higher than same architecture of conventional GAN generator, but the difference is negligible.
> - During training, generating $K$ samples is only to identify the one closest to the ground truth, and the $K-1$ unchosen samples do not retain gradients, so they are immediately discarded after sampling at the current layer, freeing up memory.
> - In the generation phase, we randomly sample one number from range($K$) as an index and only generate the sample at the chosen index, avoiding the need to generate the other $K-1$ samples, thus not occupying additional memory or computation.
> - Furthermore, in the single-shot paradigm, which is our default, a Discrete Distribution Layer (DDL) is introduced every two convolution layers. The DDL contains $K$ samples of the same size as the feature. The memory consumption of features stored in convolution layers, including conv and normalization layers, is generally significantly greater than that of $K$ samples. Similarly, the computational cost of convolution layers is significantly higher than that required to compute $K$ samples. Overall, the additional computational and memory overhead introduced by DDLs compared to conventional NN blocks is not significant.
>
> We hope this response clarifies our contributions and addresses your concerns. Thank you for considering our work.

---

> > ### Comment · Reviewer_1dyp · 2024-11-22
> > **Thanks to the authors for a detailed response**
> >
> > Thank you for your detailed response. I am skeptical of the scalability of the model due to the hierarchical approach as well as underperforming models with less parameters. However, I think that the DDN is a novel and interesting idea which can inspire further research efforts. Time will tell if this can be taken to the next level, but I think as it stands, this work is of high quality. Thus, I maintain my favorable rating.

---

> > > ### Author Response · Authors · 2024-11-23
> > > **Thanks and inquire further guidance for presentation**
> > >
> > > Thank you for your favorable rating and acknowledgment of the high quality of our work. We would also like to inquire about any improvements that could be made in terms of presentation after incorporating your feedback on the original manuscript. Currently, your presentation rating for us is 2 (fair), and we greatly appreciate any further guidance you could provide to enhance the presentation quality of our paper.

---

> > > > ### Comment · Reviewer_1dyp · 2024-11-24
> > > >
> > > > I think a discussion of the connections to VQ-VAE should be further elaborated upon. Ideally, if some fair comparison could be presented, I am willing to raise my score further.

---

> > > > > ### Author Response · Authors · 2024-11-25
> > > > > **Added connections to VQ-VAE and the FID score of VQ-VAE**
> > > > >
> > > > > Based on your suggestions, we have made the following changes in Paper revision V2:
> > > > > - Added a dedicated section in the Related Work to discuss the connections between DDN and the VQ-VAE series.
> > > > > - Included the FID score of VQ-VAE in Table 1 (FID on CIFAR10) with the data source referenced.
> > > > >
> > > > > We hope these revisions address your concerns. Thank you for considering our work.

---

> > > > > > ### Comment · Reviewer_1dyp · 2024-11-25
> > > > > >
> > > > > > Two clarification questions. 1. Is an autoregressive prior trained or how is sampling conducted for the VQ-VAE baseline? 2) In Table 1, what is meant by "The data for VQ-VAE comes from Vuong et al. (2023)?" It is CIFAR10 so why is the data not from the original source?

---

> > > > > > > ### Author Response · Authors · 2024-11-26
> > > > > > > **Reply to two clarification questions**
> > > > > > >
> > > > > > > **1. In Table 1, what is meant by "The data for VQ-VAE comes from Vuong et al. (2023)?" It is CIFAR10 so why is the data not from the original source?**
> > > > > > >
> > > > > > > We used the data from Vuong et al. (2023) [1] because we were unable to find the FID score for VQ-VAE on CIFAR10 in the following sources:
> > > > > > > - The original VQ-VAE paper conducted only reconstruction experiments on CIFAR10 and did not address generation experiments (https://arxiv.org/abs/1711.00937).
> > > > > > > - The subsequent VQ-VAE-2 paper did not involve CIFAR10 (https://arxiv.org/abs/1906.00446).
> > > > > > > - The review article on generative models (https://arxiv.org/abs/2103.04922) also does not provide the FID score for VQ-VAE in its Table 1.
> > > > > > >
> > > > > > > After several hours of searching, we finally found the FID score reported in the peer-reviewed paper by Vuong et al. (2023) [1], specifically in Table 5.
> > > > > > >
> > > > > > > Additionally, we attempted to find reproductions of VQ-VAE on CIFAR10 in the open-source community to obtain more reliable metrics. However, according to third-party reproductions, the generative quality of VQ-VAE on CIFAR10 was indeed poor, aligning with the FID score reported in [1]:
> > > > > > > - [CIFAR10 generation results](https://github.com/jiazhao97/VQ-VAE_withPixelCNNprior/raw/master/Figures/Sample_pixelcnn_cifar10.png) from https://github.com/jiazhao97/VQ-VAE_withPixelCNNprior
> > > > > > > - [CIFAR10 reconstruction results](https://github.com/nadavbh12/VQ-VAE/raw/master/images/cifar10.png) (first row: original CIFAR10 images, second row: corresponding reconstructions) from https://github.com/nadavbh12/VQ-VAE
> > > > > > >
> > > > > > > We speculate that the poor FID score of VQ-VAE might be due to the fact that in the original VQ-VAE paper, for the CIFAR experiments, the network parameters were aligned with the original VAE as much as possible, leading to limited model capacity.
> > > > > > >
> > > > > > > **2. Is an autoregressive prior trained or how is sampling conducted for the VQ-VAE baseline?**
> > > > > > > Yes, as indicated in [1] (Section 4.2.4), similar to the original VQ-VAE approach, a conventional autoregressive model, the PixelCNN, is employed to estimate a prior distribution over the discrete latent space of VQ-VAE.
> > > > > > >
> > > > > > > Thank you for considering our work.
> > > > > > >
> > > > > > > ---
> > > > > > > [1] Vector Quantized Wasserstein Auto-Encoder. Tung-Long Vuong, Trung Le, He Zhao. ICML 2023 (http://proceedings.mlr.press/v202/vuong23a/vuong23a.pdf)

---

> > > > > > > > ### Comment · Reviewer_1dyp · 2024-11-26
> > > > > > > >
> > > > > > > > Thank you for answering my two questions. As promised, I will raise my score.

---

### Official Review · Reviewer_tvs8 · 2024-11-04

**Soundness:** 2
**Presentation:** 2
**Contribution:** 2
**Rating:** 5
**Confidence:** 4

**Summary:**

The paper proposes to generate samples in a sequential way: pass the output of the previous module into a function randomly selected from a set of K functions in the current module, with the initial input to the first module being a zero vector. To train each module in the space, one first collect a trajectory with these modules except that one does not do random sampling out of K functions but pick the one with the output closet to the data sample $x$. With this trajectory, one computes L2 distances between the output of each module and $x$ and optimize it. The paper shows some empirical results on the CIFAR-10 dataset.

**Strengths:**

The proposed method is simple and seemly intuitive, and has been experimented on small-scale datasets like CIFAR-10.

**Weaknesses:**

Purely based on the presentation of the paper, I find the proposed method neither theoretically attractive, as it does not explicitly model probability distributions or model some complex distributions, nor empirically useful due to its worse performance compared to simple baselines like DCGAN. Indeed, the paper claims that VQ-VAE is unsatisfactory (e.g., in the abstract) but it does not even compared against VQ-VAE.

If we take a closer look into the proposed method, it seems to me not too different from a slightly-different VQ variant of diffusion models (e.g., https://arxiv.org/abs/2111.14822) -- running discrete diffusion (as the proposed model is basically trained in a similar way) on a latent space represented by a codebook of features, probably with some additional gradients of hierarchical latent spaces. The paper fails to connect the proposed method to a lot of existing papers and explain 1) what is different and novel given these existing methods and 2) why these design choices.

**Questions:**

See weakness.

---

> ### Author Response · Authors · 2024-11-20
> **Official Comment by Authors - (1/2)**
>
> Thank you for the thoughtful consideration of the paper and constructive feedback.
>
> **1 “does not explicitly model probability distributions”**:
> We respectfully disagree with this assessment. As described in Line #97-102, DDN explicitly approximates the target distribution by fitting the training data with a discrete distribution. The network generates multiple samples (K) simultaneously, which collectively represent a discrete distribution. Each generated sample has an equal probability mass of 1/K, and our goal is to make this discrete distribution as close as possible to the target distribution.
>
> Figure 17 (Toy examples for two-dimensional data generation) demonstrate that DDN can successfully fit various target distributions by representing them as discrete distributions composed of multiple samples.
>
> Additionally, we believe it is unfair to negatively evaluate a generative model solely because it does not explicitly model probability distributions. For instance, Generative Adversarial Networks (GANs) learn distributions implicitly without explicitly estimating probability densities.
>
> **2 “Does not model some complex distributions, nor empirically useful due to its worse performance”**:
> 1.DDN is capable of modeling complex distributions, as evidenced by its application to the FFHQ and CIFAR10 datasets, which are considered complex. Few foundational generative models can effectively model such complexity (five models from https://arxiv.org/abs/2103.04922). DDN, with its simple yet effective concept, demonstrates its capability to model these distributions effectively.
>
> 2.As other reviewers have noted:
> > "I do not think it is fair to hold this novel method to the same standard as more matured, modern approaches." -- Reviewer 1dyp
>
> > "Such papers should always be judged under the understanding that for existing methods, a lot of engineering has taken place and should be thought of as the first GAN paper or the first Diffusion models paper." -- Reviewer r4YK
>
> We emphasize that DDN is a novel generative model, representing the first work of its kind. It is not fair to compare it to more mature methods. Further research is needed to enhance DDN's performance, particularly in developing better Zero-Shot Conditional Generation models.
>
> 3.Moreover, our paper compares DDN with IGN (https://openreview.net/forum?id=XIaS66XkNA), another new generative model proposed at ICLR 2024. On the CelebA-64x64 dataset, DDN outperforms IGN in terms of FID scores and qualitative analysis (Figure 6 Line#412).
>
> 4.DDN possesses unique properties not found in other generative models, enabling novel applications. As noted by Reviewer 1dyp:
> > The ability to condition generation on external signals without gradients is a useful property.
>
> **3 “the paper claims that VQ-VAE is unsatisfactory (e.g., in the abstract) but it does not even compared against VQ-VAE.”**:
> 1. We clarify that VQ-VAE is not mentioned in our abstract or introduction. It is only discussed in the related work section, where we note its 2D latent structure and reliance on an additional prior network for generative tasks.
> 1. We were unable to find standard FID scores for VQ-VAE on CIFAR10 in several key papers, including VQ-VAE, VQ-VAE-2, and a comprehensive survey on generative models (https://arxiv.org/abs/2103.04922).
> 1. DDN is a new generative model and we prefer to compare it with other foundational generative models. VQ-VAE, being an extension of VAE requiring an additional generative model (e.g., an autoregressive model) for latent space modeling, is not a fundamental generative algorithm. Thus, we did not attempt to reproduce its results for comparison.
> 1. Our focus is on introducing a novel generative model with unique properties like a 1D latent space and Zero-Shot Conditional Generation without gradients, rather than focusing solely on performance metrics.

---

> ### Author Response · Authors · 2024-11-20
> **Official Comment by Authors - (2/2)**
>
> **4 “Similar to VQ variant of diffusion models”**:
> While both DDN and VQ-Diffusion model distributions and involve direct input/output of samples, they differ significantly:
>
> - Input/Output: VQ-Diffusion processes a single noisy sample at each step, outputting a less noisy sample. DDN, however, takes the selected output sample from the previous layer (**without noise**) and generates **multiple samples** (representing a discrete distribution).
> - Training: VQ-Diffusion computes gradients independently for each step, whereas DDN computes joint gradients across all layers and uses the "Split-and-Prune" algorithm to balance sampling probabilities.
> - Data Scope: VQ-Diffusion models only discrete data (VQ-VAE's latent), while DDN can model both discrete and continuous data.
> - Latent Space: VQ-Diffusion lacks an intuitive latent space and cannot reconstruct samples, unlike DDN, which has a 1D latent space enabling reconstruction.
> - Zero-Shot Conditional Generation: DDN supports this without gradients, which VQ-Diffusion and other models do not.
>
> **5 “fails to connect the proposed method to a lot of existing papers”**:
> Given the novelty of our method, direct connections to existing methods are limited. However, we highlight the following:
> - The approach of partitioning the large space into a hierarchical conditional probability model is inspired by autoregressive models.
> - The "Split-and-Prune" algorithm draws inspiration from evolutionary theory and genetic algorithms.
> - The technique of learning residuals is inspired by ResNet.
> - The form of Zero-Shot Conditional Generation(ZSCG) is inspired by SDEdit (https://arxiv.org/abs/2108.01073).
>
> We have further emphasized these connections in the revised version of our manuscript.
>
>
> **6 “what is different and novel”**:
> - DDN introduces a new generative modeling approach that approximates data distributions using hierarchical discrete distributions, achieved through the simultaneous generation of multiple samples.
> - It supports zero-shot conditional generation across non-pixel domains without gradients, which other models do not.
> - The latent space is a configurable tree-like structure which differ from autoregressive models.
> - It employs a novel "Split-and-Prune" algorithm to balance sampling probabilities across multiple samples.
>
> Other reviewers also acknowledge the novelty of our approach:
> > "The novelty is very strong, and this should not be overlooked. This is a whole new method, very different from any of the existing generative models." -- Reviewer r4YK
>
> > "The idea of sampling an image and autoregressively feeding it back in through each layer in a forward pass is novel to the best of my knowledge." -- Reviewer 1dyp
>
>
> **7 “why these design choices.”**:
> - **Generating multiple samples at each layer:** We posit that since network features capture distributional information, generating multiple samples simultaneously can effectively represent distributions (Line #011 in Abstract).
> - **Hierarchical conditional generating:** Current neural networks cannot generate a vast number of samples simultaneously. We adopt a strategy from autoregressive models to partition this space into a hierarchical conditional probability model (Line #105 in Introduction).
> - **Split-and-Prune:** We discuss the necessity of this algorithm to balance sampling probabilities across multiple samples in Section 3.1 Optimization with Split-and-Prune (Line #252).
>
> We hope this response clarifies our contributions and addresses your concerns. Thank you for considering our work.

---

> ### Comment · Reviewer_tvs8 · 2024-11-23
>
> Thanks for the authors' response. While I appreciate the comprehensive explanation on the motivation and technical details, I am not convinced by the authors' arguments.
>
> ### Why DDN is just some special kind of hierarchical VQ-VAE (specifically, VQ-VAE-2)
>
> **During inference time:** For each layer in DDN, the sampling procedure is essentially (for each layer) uniformly sampling a random "feature" in the codebook, thus exactly the same as a hierarchical VQ-VAE (with less efficient encoding as one stores the latent code of the full image, instead of patches).
>
> **During training time:** the $i$-th layer searches for the "code" in a codebook of size $K$ that is nearest to its input, again a standard approach in VQ-VAE. Indeed, we obtain DDN if we make the following modifications to VQ-VAE-2 (Figure 2a in [1]):
>
> - The latent dimension is of the same size of the inputs.
> - The encoders in the stacked VQ-VAEs set to always output zero.
>
> I do agree that this is something not traditional, and therefore I strongly believe that the authors need to provide more theoretical justifications and motivations for it. More importantly, I have a strong feeling that the authors overlook some of the existing literature and do give enough credits to the previous work in this field.
>
> ### The so-called "split-and-prune" approach is not novel
>
> The "nodes" are essentially codes in a codebook, if we adopt the terminology in VAE literature. It is nothing new that people will re-initialize some of the "dead" codes near highly used codes (for instance, see Sec 3.1 in this paper [2], or Sec 5.3 in [3]).
>
> ### On evaluation
>
> First, FID should not be too hard to compute if one already has a trained VQ-VAE model.
>
> Second, if you sample in a way like VQ-VAE (a uniform sampling of latent codes), you should be able to compute the negative log-likelihood metric, as shown in the original VQ-VAE paper.
>
> **Conclusion.** I believe that the authors are not well aware of the literature of the field and they fail to acknowledge the great similarities between the proposed DDN and existing approaches, as DDN can simply be seen as a special VQ-VAE-2. In addition, both thorough theoretical analysis and sufficient empirical results are missing to justify the specific design choice. I strongly believe that the current form of this paper is not ready for the ICLR conference.
>
> ---
> [1] Generating Diverse High-Fidelity Images with VQ-VAE-2. Ali Razavi, Aaron van den Oord, Oriol Vinyals. NeurIPS 2019.
>
> [2] Jukebox: A Generative Model for Music. Prafulla Dhariwal, Heewoo Jun, Christine Payne, Jong Wook Kim, Alec Radford, Ilya Sutskever. https://arxiv.org/abs/2005.00341
>
> [3] Hierarchical Quantized Autoencoders. Will Williams, Sam Ringer, Tom Ash, John Hughes, David MacLeod, Jamie Dougherty. NeurIPS 2020

---

> > ### Comment · Reviewer_r4YK · 2024-11-24
> > **Adding some points about VQ-VAE relation to the discussion**
> >
> > I want to thank Reviewer tvs8 for their insights.
> > While there are strong relations between DDN and VQ-VAE, I respectfully disagree regarding DDN being "some special kind" of them. I think there is a pretty solid distance between them and I thought it is a good idea to share my opinion with Reviewer tvs8. If I am inaccurate, the authors are welcome to correct me.
> >
> > I would like to point to a major difference, other than some that were pointed out before.
> > VQ-VAE encodes the image to an array of spatial tokens. DDN applies nearest neighbor over an entire image at each layer, making the code the route that was chosen along the nodes of the tree. While both implicitly or explicitly use nearest neighbors to a dictionary, they are crucially different.
> >
> > VQ-VAE variants have a second training phase, which is training some autoregressive strategy to predict the image tokens. This is typically done by autoregressive generative model, e.g., pixel-RNN in the basic version or a transformer with highest confidence masked tokens in Muse (Chang et al.).
> > DDN has no such second phase.
> >
> > Then at inference, VQ-VAE variants apply their trained autoregressive strategy predicting the image tokens and then decoding them. This suggests that the coding itself is not enough for the generative model to hold. If one tries to sample random tokens, the decoded result will not look like anything. Some of the generative process is the token prediction, which means the latent space is not the set of possible combinations of tokens, but implicitly embedded in the stochasticness of the autoregressive generative model.  DDN, as far as I understand ideally always outputs valid results, no combination of nodes is out-of-distribution and there is no need for an autoregressive generative model to choose the route. This means the entire generative coding is this 1d latent.
> >
> > I think these are crucial differences- spatial tokens vs route of nodes, additional generative mechanism vs a holistic model.
> > I hope I could contribute to this discussion,
> > Thanks again to Reviewer tvs8 and the authors.

---

> > ### Author Response · Authors · 2024-11-27
> > **Official Comment by Authors - Round 2 - (1/2)**
> >
> > We thank Reviewer tvs8 for the detailed feedback and appreciate the constructive discussions from reviewers r4YK and 1dyp. Below, we address the similarities and differences between DDN and VQ-VAE-2 and related methods.
> >
> > **8 “just some special kind of hierarchical VQ-VAE (specifically, VQ-VAE-2)”**
> > Even if we base our discussion on your assumption, there is a crucial oversight. The multiple samples simultaneously output by each layer of DDN are fundamentally different from the codebook in VQ-VAE. One represents features, while the other represents static embedding parameters. This distinction leads to significant issues, as detailed below:
> >
> > In VQ-VAE, the codebook consists of a set of independent parameters (also known as embeddings) that can be optimized by the loss function. These parameters are solely influenced by the loss and cannot be affected by the input, unlike features. During forward propagation, the input image only determines which embedding is selected, without altering the embedding's value itself.
> >
> > If I understand correctly, in your hypothesized VQ-VAE-2, each embedding is as large as the training samples, and the embedding closest to the current training sample at each layer is chosen and fed into the decoder to generate the image. Additionally, each layer has a Vector Quantization loss that forces the selected embedding to fit the current ground truth (GT) sample (as shown in Equation (3) of the original VQ-VAE paper).
> >
> > This leads to the following issues:
> > - Lack of Correlation Across Layer Dimensions: VQ-VAE-2 feeds all selected embeddings from each layer into the decoder to obtain a single output. After selecting the embedding closest to the current GT sample at each layer, it is directly input into the decoder without any hierarchical output. Consequently, there is no conditioning of one layer's output on the next, leading to independence among layers and a lack of correlation.
> > - Homogeneity of Embeddings Across Layer Dimensions: Due to the absence of an encoder, the inputs to the embeddings at different layers are identical, i.e., the current GT sample. Unlike the original VQ-VAE with an encoder, where features at different layers are inherently different. The uniform input and identical optimization objective (fitting the current GT sample using Vector Quantization loss) result in homogeneous embeddings across layers. These embeddings represent an average fit to the entire dataset, lacking complementarity and synergy.
> > - Lack of Diversity in Sample Dimensions: Since embeddings are fixed parameters, different GT samples only influence which embedding is selected, without altering the embedding's value. This limitation means that each layer can only choose from a limited set of $K$ static embeddings, resulting in a lack of diversity. Moreover, these selected embeddings represent an "average fit to the entire dataset" and cannot effectively represent the current GT sample.
> >
> > Combining these embeddings, which are independent across layer dimensions, lack correlation, and exhibit severe homogeneity, with a limited overall sample space (only $K$ fixed embeddings per layer, regardless of GT sample variations), makes it challenging for the decoder to reconstruct the target sample.
> >
> > In contrast, the multiple samples output by each layer of DDN are features generated by the network, not static embeddings, allowing them to dynamically change based on the input. Furthermore, conditioned on the selected sample from the previous layer, the generated samples resemble the selected sample while exhibiting slight differences among themselves. As the number of layers increases, the generated images become increasingly similar to the current GT sample (as illustrated in Figure 1(a) of the main text).
> >
> > Moreover, we believe that the assumption itself is unreasonable:
> > - For VQ-VAE, both the encoder and the second stage (prior model) are crucial. No existing work suggests that these components can be simply removed.
> > - Under your assumption, the embeddings would occupy a substantial amount of space, with a total dimensionality of $W \times H \times C \times K \times L$ (width, height, channels, codebook size, layers). Even in our simplest MNIST experiment, the embedding space reaches $28 \times 28 \times 1 \times 64 \times 10 = 490K$, exceeding the total network parameters of DDN (407K). Since embeddings are part of the network parameters, this significantly reduces the efficiency of parameter utilization.
> >
> > Other key differences include:
> > - DDN supports Zero-Shot Conditional Generation (ZSCG), whereas the hypothesized VQ-VAE-2 has only a final output without a refinement process, making ZSCG infeasible.
> > - The hypothesized VQ-VAE-2 cannot achieve the Recurrence Iteration Paradigm demonstrated in Figure 3(c) because each layer only has a sampler for finding the nearest neighbor, without the ability to input the generated sample from the previous layer as a condition.

---

> > > ### Author Response · Authors · 2024-11-27
> > > **Official Comment by Authors - Round 2 - (2/2)**
> > >
> > > **9 “The "Split-and-Prune" approach is not novel”**
> > > - We greatly appreciate your observation regarding the similarity between the "Split-and-Prune" algorithm and the strategy of re-initializing some of the "dead" codes near highly used codes. We concur with your viewpoint.
> > > - In revision V2, we acknowledged this similarity in the Related Work section and Section 3.1 (Optimization with split-and-prune), providing corresponding explanations and references.
> > >
> > >
> > > **10 “On evaluation, FID should not be too hard to compute if one already has a trained VQ-VAE model.”**
> > > After extensive searching, we finally found FID scores for VQ-VAE on CIFAR10 reported in Table 5 of a peer-reviewed paper [1]. We have included this information in Table 1 (FID on CIFAR10) of revision V2.
> > >
> > > ---
> > > [1] Vector Quantized Wasserstein Auto-Encoder. Tung-Long Vuong, Trung Le, He Zhao. ICML 2023 (http://proceedings.mlr.press/v202/vuong23a/vuong23a.pdf)
> > >
> > >
> > >
> > > **11 “overlook some of the existing literature and do give enough credits to the previous work in this field.”**
> > > **“severely underestimates the connections between DDN and existing approaches.”**
> > > **“I feel hard to ignore the similarities there and to not spend words discussing them (VQ-VAE).”**
> > > In revision V2, we made the following changes to enhance the connections with existing methods:
> > > - Added a dedicated paragraph in the Related Work section to discuss the connections between DDN and the VQ-VAE series.
> > > - Acknowledged the similarity between Split-and-Prune and the strategies used to address codebook collapse in VQ-VAE in both the Related Work section and Section 3.1 (Optimization with split-and-prune).
> > >
> > >
> > > **12 “I believe that the current form of this paper is not ready for ICLR.”**
> > > We would like to emphasize our main contributions:
> > > - We propose a novel generative modeling approach that approximates target distributions (including continuous distributions) using discrete distributions, demonstrating its effectiveness on CIFAR10 and FFHQ.
> > > - DDN possesses a unique property of supporting Zero-Shot Conditional Generation without gradients, making it the first generative model with this capability.
> > >
> > > Additionally, IGN [2], another similar work proposing a new generative model algorithm, was published in ICLR 2024. On the CelebA-64x64 dataset, DDN outperforms IGN in both FID scores and qualitative analysis (as shown in Figure 6 in the main text).
> > >
> > > [2] Idempotent Generative Network. Assaf Shocher, Amil Dravid, Yossi Gandelsman. In ICLR, 2024. (https://openreview.net/forum?id=XIaS66XkNA)
> > >
> > > ---
> > >
> > > We hope this response clarifies our contributions and addresses your concerns. Thank you for considering our work.

---

> > > > ### Author Response · Authors · 2024-12-01
> > > > **Friendly Reminder**
> > > >
> > > > Dear Reviewer tvs8:
> > > >
> > > > As we approach the end of the discussion period, we would like to know whether our revision and response have addressed your concerns to merit an increase in the rating, or if there are any remaining issues that you would like us to clarify.
> > > >
> > > > Thank you again for your time and effort.

---

> > > > > ### Comment · Reviewer_tvs8 · 2024-12-03
> > > > >
> > > > > I appreciate the authors' time and efforts in refining their arguments, presentations and experiments. Some parts of my concerns are resolved, while I am less satisfied with the theoretical aspects in the proposed method. I still believe that more connections need to be built between related methods (VQ-VAE and discrete diffusions), which deserve a full discussion section to elaborate on. I also believe that the authors should further lower their tone in claiming the novelty in the split-and-prune strategy, as I see few differences between theirs and the existing ones.
> > > > >
> > > > > What I do agree with other reviewers: it is worth encouraging exploration in other directions in building models and architectures for generative models. I would therefore refrain from strongly rejecting the paper and raise my rating to 5 and probably leave the decision to the ACs, though I feel that there is still much room for the paper to improve (in theoretical analysis/modeling, in discussions on connections to related work, and in paper presentation). I personally believe taking one or two months of paper refinement for another major conference sounds like a better option to me. If the paper is eventually accepted, I strongly encourage the authors to devote significant amount of time on these aspects.

---

> > > > > > ### Author Response · Authors · 2024-12-03
> > > > > > **Official Comment by Authors**
> > > > > >
> > > > > > We greatly appreciate your constructive feedback and will further refine our paper in future versions based on your suggestions.
> > > > > >
> > > > > > Thank you once again for your time and effort.

---

> ### Comment · Reviewer_tvs8 · 2024-11-24
>
> I appreciate Reviewer r4YK's for sharing their opinions. While I agree that this is a different design choice, conceptually I feel that it still lies in the domain of VQ-VAE-like models, as one could simply discard the second stage in VQ-VAE-2 and do random uniform sampling. From my perspective, the differences are:
>
> - No encoder (as mentioned in my earlier response to the authors)
> - Depth of the hierarchy. VQ-VAE-2 has two stages while DDN does more.
> - Token size ("width"). We may have a spectrum of token sizes, from each pixel as a token to the whole image an a token. VQ-VAE picks token size to be 4x4 while in DDN it is the whole image.
>
> In my opinion, these design choices do not constitute a brand new model. Here I would like to make an analogy between ViT and many other previous attempts that directly model pixels as tokens -- how the authors of ViT claims their contribution is that they find this surprisingly good design choice and empirically justify the story. In the paper of DDN, I do not see good empirical results, nor very convincing theoretical modeling.
>
> As a result, I strongly believe that the way the authors present their ideas and results 1) does not thoroughly show the benefits and trade-offs between there design choices, and more importantly, 2) severely underestimates the connections between DDN and existing approaches. It also remains to be resolved for the previously mentioned issues, *e.g.* the novelty claim on the "split-and-prune" strategy. Given this short period of time, these issues are unlikely to be fixed for ICLR.
>
> That's said, I do feel that it is something interesting and worth investigating (both theoretically and empirically) to trade "width" with "depth" in VQ-VAE models, if no other paper has shown similar things. Less width means that one does not need to keep track of the distribution in the spatial domain but rather rely on the depth of the hierarchy. I am curious to see how VQ-VAE-2 without the second stage (so doing random sampling at inference time) behaves on simple datasets, and I would like to see some ablation study to see how random sampling on the spatial domain works for different "depths" and different "widths" and to see at which point random sampling will fail. There are many other aspects one could inspect for sure. And on the story of routing and compression: it is also interesting to some degree if the authors could do more analysis and building deeper connections to other theories. Therefore, I encourage the authors to make a major revision of their paper on storytelling, presentation, theoretical analysis and empirical experiments.

---

> ### Comment · Reviewer_r4YK · 2024-11-24
>
> Thank you for maintaining an active discussion, I appreciate that.
>
> Don't you think that classifying a method on images as a method for patches with a design choice of one huge patch is overgeneralization? I feel it's a bit like claiming that $x^2$ is a special case of linear function $ax$ with a design choice of $a=x$. The spatial token approach is some sort of generative inpainting, gradually filling up masked tokens. This is a very spatial approach in nature and hard to see it generalizes to data that is not spatial or sequential. The proposed DDN has no spatial notion at all and potentially could be adjusted to any type of input.
>
> Random sampling instead of second-stage autoregressive token prediction with VQ-VAE does not produce reasonable results. This is a known fact, and this is the reason for the existence of that stage 2. Making the token prediction faster was a research direction that took place ~2 years ago. Again, this is very different from the proposed DDN that has no 2nd stage and ideally only produces valid outputs.
>
> Thanks again for the discussion.

---

> ### Comment · Reviewer_tvs8 · 2024-11-24
>
> Thank you for your reply and I am glad to hear your opinions.
>
> Most VQ-VAE variants indeed use spatial tokens. Yet, I believe that VQ-VAE is a generic method that does not only apply to images -- the idea of "using VQ in the latent space + training an autoencoder" is widely applicable to many other domains. I was thinking that it was only because people aimed to develop models for images that spatial tokens were introduced (especially considering the existence of competitor models like GANs).
>
> On the autoregressive 2nd-stage: I am less knowledgeable in this specific technical detail, but I do wonder what if we use large enough spatial token in VQ-VAE-2 (provided that we disable the encoders). The interpolation between DDN and VQ-VAE-2 seems very easy and intuitive to me. I do agree that VQ-VAE-2 was proposed for some other reasons, but if I were the authors of DDN, I feel hard to ignore the similarities there and to not spend words discussing them.
>
> Even though we assume that DDN is of reasonable distance away from existing models, what concerns me most is that the lack of connections to existing models and lack of theoretical analysis (not saying that the authors need to prove some bound but at least something more concrete). I do not tend to argue very strongly against DDN's novelty (except for the "split-and-prune" thing), as I find Reviewer r4YK probably better in estimating the contribution in this specific domain. I feel a bit hard to change my stance if the rest of the issues are not fixed or if there are not some other strong reasons.

---

> > ### Comment · Reviewer_1dyp · 2024-11-24
> >
> > In my opinion,  I see sufficient differences between DDN and VQ-VAE. The biggest non-trivial differences I see are 1) no second stage training of an autoregressive prior 2) nearest neighbor look up on the entire image. I also think that the applications that DDN enables, the ZSCG, are interesting. I agree that some connections to VQ-VAE should be elaborated upon in the manuscript. However, I feel the paper should still be accepted.

---

### Author Response · Authors · 2024-11-20
**Paper revision V1**

We have updated the draft based on the reviewers' comments. All modifications and additions are **highlighted in yellow**. Here is a summary of the changes:

1. Figure 5 (2-D Toy examples) has been moved to the Appendix. Note that this adjustment has resulted in corresponding changes to the layout and figure indices throughout the manuscript.
2. Relocated Table 2 to be adjacent to Section 4.2, the Ablation Study, for better readability and contextual relevance.
3. Further emphasized the connections to existing papers.
4. Clarified that the Single Shot Generator paradigm is our default choice for most experiments.

---

### Author Response · Authors · 2024-11-25
**Paper revision V2**

We have updated the draft based on the reviewers' new replies. All modifications and additions are highlighted in yellow. Here is a summary of the changes:
- Added a dedicated section in the Related Work to discuss the connections between DDN and the VQ-VAE series.
- Included the FID score of VQ-VAE in Table 1 (FID on CIFAR10) with the data source referenced (from Table 5 in [1]).
- Acknowledged the similarity between Split-and-Prune algorithm and the strategies used to address codebook collapse in VQ-VAE in the Related Work and Section 3.1 (Optimization with split-and-prune)

---
[1] Vector Quantized Wasserstein Auto-Encoder. Tung-Long Vuong, Trung Le, He Zhao. ICML 2023 (http://proceedings.mlr.press/v202/vuong23a/vuong23a.pdf)

---

### Meta-Review · Area_Chair_HaUk · 2024-12-19

**Metareview:**

This paper suggests the Discrete Distribution Network (DDN) as a new generative model paradigm. The DDN is composed of Discrete Distribution Layers (DDLs) that each generate several images out of its input feature. For training the closest generated image to the training sample is chosen and compared to; for conditional sampling, different conditioning mechanisms are used to select the image in each layer. Two reviewers found this idea novel while the third reviewer found the method close to the VQ-VAE method. Following a long and detailed discussion among reviewers it seems that although the proposed method indeed has some similarities to VQ-VAE (closest neighbor discrete samples) it can be considered sufficiently different/novel due to its single stage training/sampling, non-static and global codes/features, and autoregressive image generation across DDLs. Other concerns include scalability of the method, theoretical justification, and limited experimental setup but reviewers agree that for a new generative paradigm the paper contains sufficient evidence for its merit.

**Additional Comments On Reviewer Discussion:**

No additional comments.

---

### Decision · Program_Chairs · 2025-01-22

Accept (Poster)